# ◎ DART-Math: Difficulty-Aware Rejection Tuning for Mathematical Problem-Solving

**Yuxuan Tong**[*1]**, Xiwen Zhang**[2]**, Rui Wang**[2]**, Ruidong Wu**[2]**, Junxian He**[3]

[1]Tsinghua University    [2]Helixon Research    [3]HKUST

tongyx21@mails.tsinghua.edu.cn    junxianh@cse.ust.hk

## Abstract

Solving mathematical problems requires advanced reasoning abilities and presents notable challenges for large language models. Previous works usually synthesize data from proprietary models to augment existing datasets, followed by instruction tuning to achieve top-tier results. However, our analysis of these datasets reveals severe biases towards easy queries, with frequent failures to generate any correct response for the most challenging queries. Hypothesizing that difficult queries are crucial to learning complex reasoning, we propose *Difficulty-Aware Rejection Tuning* (DART), a method that allocates difficult queries more trials during the synthesis phase, enabling more extensive training on difficult samples. Utilizing DART, we have created new datasets for mathematical problem-solving that focus more on difficult queries and are substantially smaller than previous ones. Remarkably, our synthesis process solely relies on a 7B-sized open-weight model, without reliance on the commonly used proprietary GPT-4. We fine-tune various base models on our datasets ranging from 7B to 70B in size, resulting in a series of strong models called DART-Math. In comprehensive in-domain and out-of-domain evaluation on 6 mathematical benchmarks, DART-Math outperforms vanilla rejection tuning significantly, being superior or comparable to previous arts, despite using much smaller datasets and no proprietary models. Furthermore, our results position our synthetic datasets as the most effective and cost-efficient publicly available resources for advancing mathematical problem-solving.[1]

## 1 Introduction

Recent years have seen remarkable advancements in various tasks through the use of large language models (LLMs) (Brown et al., 2020; Touvron et al., 2023; Chowdhery et al., 2023; Anthropic, 2023; OpenAI et al., 2023). However, these models still struggle with complex reasoning (Hendrycks et al., 2021; Jimenez et al., 2024; He et al., 2024; Lin et al., 2024), a cornerstone of human cognitive essential for tackling intricate tasks. Mathematical reasoning, in particular, represents a significant challenge and stands as one of the most difficult categories of reasoning for state-of-the-art LLMs (Hendrycks et al., 2021; Cobbe et al., 2021b; Zheng et al., 2022).

In this work, we focus on mathematical problem-solving to explore enhancement of the mathematical reasoning abilities of pretrained LLMs. We investigate instruction tuning (Longpre et al., 2023; Wang et al., 2023), which is recognized as the most cost-effective method and achieves the state-of-the-art performance on various mathematical benchmarks (Yu et al., 2024; Yue et al., 2024). Current SOTA instruction tuning methods for mathematical problem-solving are typically implemented as

---

[*]Work done during visit to HKUST.

[1]Our datasets, models and code are publicly available at https://github.com/hkust-nlp/dart-math.

38th Conference on Neural Information Processing Systems (NeurIPS 2024).

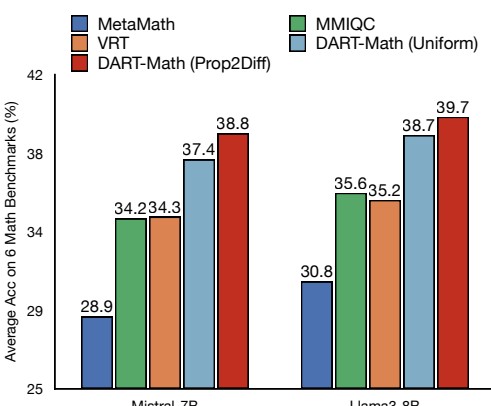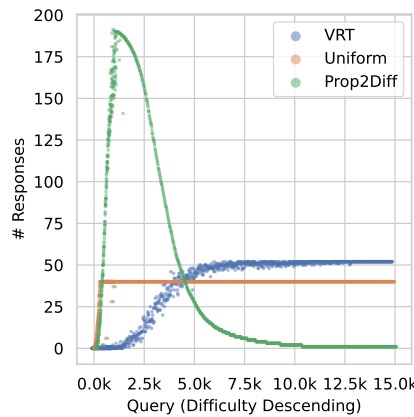

Figure 1: **Left:** Average accuracy on six mathematical benchmarks. We compare with models fine-tuned on the best, public instruction tuning datasets for mathematical problem-solving: MetaMath (Yu et al., 2024) with 395k examples, MMIQC (Liu et al., 2024a) with 2.3 million examples, as well as vanilla rejection tuning (VRT) with 590k examples. Both DART-Math (Uniform) and DART-Math (Prop2Diff) use 590k training examples. **Right:** Number of responses for each query descending by difficulty across 3 synthesis strategies. Queries are from the MATH training split (Hendrycks et al., 2021). VRT is the baseline biased towards easy queries, while Uniform and Prop2Diff are proposed in this work to balance and bias towards difficult queries respectively. Points are slightly shifted and downsampled for clarity.

augmenting existing training datasets with synthetic data generated from proprietary models like GPT-4 (OpenAI et al., 2023). A prevalent method of data augmentation is to sample multiple responses to given queries from a strong model and filter out the incorrect ones. This method, known as rejection tuning, ensures the high quality of the augmented thought steps and yields competitive performance (Yuan et al., 2023; Yu et al., 2024; Singh et al., 2023).

However, after careful examination of these SOTA synthetic datasets, we find that they suffer from a severe bias towards responses to easy queries and low coverage for hard queries. For example, as shown in Figure 2 (Left and Middle), while the original queries vary in difficulty, the augmented samples in the MetaMathQA dataset (Yu et al., 2024) focus more on easier queries, with zero new responses generated for 51.1% of the most difficult training queries in the MATH training set (Hendrycks et al., 2021). This phenomenon commonly exists in rejection-sampling-based data synthesis which typically samples *an equal number of raw responses for each query*, disadvantaging difficult queries that are less likely to yield correct responses. We hypothesize that such biases hinder the learning of mathematical problem-solving, since difficult examples are often deemed more crucial during training (Sorscher et al., 2022; Burns et al., 2023; Liu et al., 2024b).

To address this issue, we propose *Difficulty-Aware Rejecting Tuning* (DART), a method that prioritizes more sampling trials for challenging queries, thereby generating synthetic datasets enriched with more responses for difficult questions compared to previous methods. Specifically, we develop two strategies to achieve this: *Uniform* which collects the same number of correct responses for all queries, and *Prop2Diff* which biases the data samples towards the difficult queries, contrasting with vanilla rejection tuning. These different strategies are summarized in Figure 1 (Right), where the difficulty of a query is automatically assessed by sampling multiple responses and calculating the ratio of incorrect answers. Our difficulty-aware synthesis produces two synthetic datasets corresponding to Uniform and Prop2Diff strategies respectively, consisting of ∼590k examples. Notably, while previous works mostly utilize GPT-4 to synthesize data, we only rely on the DeepSeekMath-7B-RL model (Shao et al., 2024) to produce all the data, thereby eliminating dependence on proprietary models.

In our experiments, we evaluate DART based on Mistral-7B (Jiang et al., 2023), DeepSeekMath-7B (Shao et al., 2024), Llama3-8B, and Llama3-70B (Meta, 2024), creating a series of strong mathematical models that termed DART-Math. Across 6 in-domain and challenging out-of-domain benchmarks, DART-Math significantly outperforms vanilla rejection tuning and the baselines trained on the previously established top public datasets as shown in Figure 1 (Left), this is often achieved with smaller training data size. For example, DART-Math improves Llama3-8B from 21.2% to 46.6% on MATH (Hendrycks et al., 2021), and from 51.0% to 82.5% on GSM8K (Cobbe et al., 2021a); Our results mark the DART-Math datasets as the state-of-the-art *public* resources of instruction tuning for mathematical problem-solving.

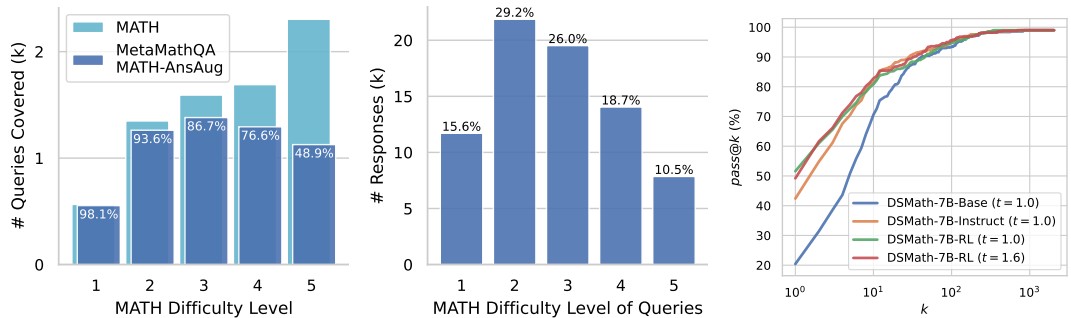

Figure 2: **Left:** Number of queries in the MATH training set and the MetaMathQA-MATH-AnsAug set across 5 difficulty levels annotated by humans. MetaMathQA-MATH-AnsAug is generated through rejection sampling from the original training queries. We annotate the query coverage ratio of MetaMathQA. While the most difficult queries (Level 5) are predominant in the original set, synthetic examples bias towards easier queries, dropping over 50% of the most difficult queries. **Middle:** Total number of responses for queries across different difficulty levels in MetaMathQA-MATH-AnsAug. The most difficult queries represent the smallest proportion, only accounting for 10.5% of all the samples. **Right:** $pass@k$ accuracy of different DeepSeekMath (DSMath) models and temperatures ($t$) on MATH500 (Lightman et al., 2024), a subset of MATH test set. With enough trials, models are actually able to sample out answer-correct responses to most (>99%) queries.

## 2 Biases in Rejection-Based Data Synthesis

In this section, we first introduce the background for rejection sampling and rejection tuning, and then present our examination on the biases of rejection-based data synthesis.

### 2.1 Background: Rejection Sampling and Rejection Tuning

We begin by formulating the data synthesis setting used for instruction tuning. For instruction tuning, the training dataset consists of $(x, y)$ pairs, where $x$ is the input query and $y$ is the response. The process of data synthesis involves generating new $(x, y)$ pairs to augment the original training dataset, thereby enhancing performance. For each input query $x_i$, it is typical to sample $M$ responses from advanced models such as GPT-4, forming the set $\{(x_i, y_i^{(j)})\}_{j=1}^M$. In the context of mathematical problem-solving, a subsequent filtering step is often implemented to eliminate incorrect $y_i^{(j)}$. This elimination is based on whether the final answer in the synthetic response aligns with the ground-truth answer.[2] This is crucial as mathematical reasoning poses a significant challenge for current LLMs, and the generated $y_i^{(j)}$ may often be of poor quality. This method of response sampling is known as *rejection sampling*, and the subsequent fine-tuning process is referred to as *rejection tuning*, which is widely employed to enhance the mathematical problem-solving abilities of LLMs (Zelikman et al., 2022; Yuan et al., 2023; Yu et al., 2024; Singh et al., 2023; Xu et al., 2024). In addition to response synthesis, the queries are typically kept constant (Singh et al., 2023; Hosseini et al., 2024; Toshniwal et al., 2024) or altered in a controlled manner (Yu et al., 2024) to ensure that ground-truth answers are readily available, which facilitates the implementation of rejection sampling. While some studies also synthesize queries without utilizing rejection tuning (Li et al., 2024; Tang et al., 2024), our focus in this work is primarily on rejection tuning, a method prevalently used for advancing the mathematical skills of LLMs.

### 2.2 On the Imbalance of Rejection-Based Data Synthesis

Next, we examine a representative synthetic dataset to identify the inherent biases present in rejection-based data synthesis as implemented in most existing works. Specifically, our analysis focuses on the AnsAug subset of the MetaMathQA-MATH dataset (Yu et al., 2024), which is a synthetic dataset that produces multiple responses for each query in the original training set of the MATH dataset (Hendrycks et al., 2021), through rejection sampling as described in §2.1. MetaMathQA has been recognized as one of the most effective synthetic datasets for mathematical problem-solving.

---

[2]Strictly speaking, final answer correctness does not necessarily imply intermediate reasoning correctness. We do not make further distinction across this paper which is not our focus.

We concentrate on the MATH split because it is a notably challenging benchmark in mathematical reasoning, equipped with human-annotated difficulty levels that aid in our analysis.

**Rejection-based data synthesis biases towards easy queries:**   Across different difficulty levels, Figure 2 (Left) shows the original query distribution of the MATH training dataset as well as the new query distribution after synthesis in the MetaMathQA-Math dataset. While the most difficult queries (Level 5) takes the largest proportion in the original query set, MetaMathQA changes the query distribution implicitly towards easier queries, dropping many hard problems. For instance, the proportion of Level 5 (the most difficult) queries notably decreases by 51.1%, indicating that rejection sampling fails to generate any correct response for those queries. As a result, as depicted in Figure 2 (Middle), the responses to the most difficult queries only account for 10.5% of all the samples. Such a phenomenon generally exists in datasets synthesized through the conventional rejection sampling method outlined in §2.1, primarily because *the same number of responses* is sampled for each query, yet the likelihood of obtaining correct responses for difficult queries is significantly lower, sometimes even zero. We hypothesize that this bias towards easy queries could substantially undermine the effectiveness of instruction tuning, as hard queries are often considered critical for instruction tuning (Lu et al., 2024; Liu et al., 2024b). We note that this bias towards easy queries is less pronounced on relatively simple datasets such as GSM8K (Cobbe et al., 2021a), where most queries are easier and it is not difficult to sample correct responses for most of the queries. However, the bias remains a significant concern when tackling challenging tasks, which represent a more compelling and complex field of study for LLMs. Building on these findings, we will next introduce our method as a potential remedy to the limitations of vanilla rejection tuning.

## 3  DART — Difficulty-Aware Rejection Tuning

### 3.1  Open-Weight Models Are Able to Generate Good Responses

Intuitively, we aim to collect a sufficient number of responses for the difficult queries. To assess whether this goal is achievable, given that models might not generate correct responses for challenging queries despite extensive sampling, we explore the capabilities of DeepSeekMath-7B-RL (Shao et al., 2024), a strong model specifically trained for mathematical reasoning. Figure 2 (Right) demonstrates the $pass@k$ accuracy on the queries in MATH500 (Lightman et al., 2024), a subset of MATH test set, indicating the proportion of queries that have at least one correct response when sampling $k$ responses for each query. Notably, even though the synthesis model possesses only 7B parameters, a 90% $pass@k$ accuracy can be achieved when sampling over 100 responses per query. These results are consistent with the findings from recent studies (Toshniwal et al., 2024; Shao et al., 2024; Li et al., 2024), which suggest that strong open-weight models are able to synthesize correct responses for most of the queries. This evidence supports the potential for effectively mitigating the insufficient coverage for difficult queries through strategic response sampling, which we introduce next.

### 3.2  DARS — Difficulty-Aware Rejection Sampling

Motivated by the observation above, we aim to collect more responses for harder queries. Specifically, we introduce two strategies to increase the number of correct responses for difficult queries: (1) **Uniform**, which involves sampling responses for each query until each query accumulates $k_u$ correct responses, and $k_u$ is a preset hyperparameter determined by the desired size of the synthetic dataset; (2) **Prop2Diff**, where we continue sampling responses until the number of correct responses for each query is (linearly) proportional to its difficulty score. The most challenging queries will receive $k_p$ responses and $k_p$ is a hyperparameter. This method introduces a deliberate bias in the opposite direction to vanilla rejection sampling, towards more difficult queries. Prop2Diff is inspired by previous works that demonstrate difficult queries can be more effective to enhance model capabilities (Sorscher et al., 2022; Liu et al., 2024b). Both the Uniform and Prop2Diff strategies prescribe a specific number of correct response for each query, determined by $k_u$ or $k_p$. Nevertheless, there are certain queries which we cannot sample out the designated number of correct responses even with extensive sampling efforts. To avoid endless running of the synthesis, we impose a cap on the maximum allowable number of raw samples per query as $n_{\max}$ — once this limit is reached for a particular query, we cease further sampling and retain any correct responses that have been gathered. The straightforward implementation of the Prop2Diff strategy risks generating no synthetic responses for easier queries if $k_p$ is set small. To mitigate this, we guarantee at least one synthetic response for

| Dataset | # Samples (k) | Synthesis Agent | Open-Source |
|---|---|---|---|
| WizardMath (Luo et al., 2023) | 96 | GPT-4 | ✗ |
| MetaMathQA (Yu et al., 2024) | 395 | GPT-3.5 | ✓ |
| MMIQC (Liu et al., 2024a) | 2294 | GPT-4+GPT-3.5+Human | ✓ |
| Orca-Math (Mitra et al., 2024) | 200 | GPT-4 | ✓ |
| Xwin-Math-V1.1 (Li et al., 2024) | 1440 | GPT-4 | ✗ |
| KPMath-Plus (Huang et al., 2024) | 1576 | GPT-4 | ✗ |
| MathScaleQA (Tang et al., 2024) | 2021 | GPT-3.5+Human | ✗ |
| DART-Math-Uniform | 591 | DeepSeekMath-7B-RL | ✓ |
| DART-Math-Hard | 585 | DeepSeekMath-7B-RL | ✓ |

Table 1: Comparison between our DART-Math datasets and previous mathematical instruction tuning datasets. Most of previous datasets are constructed with ChatGPT, and many of them are not open-source, especially for ones of the best performance.

each query when implementing Prop2Diff. While it might seem sufficient to rely on the original, real training dataset to ensure at least one human-annotated response per query, our findings highlight the importance of maintaining synthetic response coverage to learn to solve easy problems, as we will quantitatively shown in §4.3, partially because the human-annotated response is less detailed and not as beneficial as synthetic responses, demonstrated previously in Yu et al. (2024). For both Uniform and Prop2Diff strategies, we use the DeepSeekMath-7B-RL model to synthesize responses. We refer to the two sampling strategies as DARS-Uniform and DARS-Prop2Diff respectively. Though most previous methods are difficulty-agnostic, a few methods try assigning more budget to more complex questions to boost coverage, such as ToRA (Gou et al., 2024) and MARIO (Liao et al., 2024). However, ToRA/MARIO mainly focus on improving coverage without managing the distribution explicitly, leading to datasets that may still bias towards easy queries, while DARS explicitly controls the final distribution of the training dataset, completely eliminating the bias and also achieving higher coverage on the hardest queries. For more details about the comparison, we refer readers to Appendix A. As DARS-Prop2Diff requires assessing difficulties of queries, next we introduce an automatic approach to measure difficulties.

**Evaluating Difficulty:** Previous studies have used proprietary models like ChatGPT to assess the difficulty or complexity of data samples (Lu et al., 2024; Liu et al., 2024b). In this work, we introduce a new metric, *fail rate* — the proportion of incorrect responses when sampling $n_d$ responses for a given query — as a proxy for difficulty. This metric aligns with the intuition that harder queries less frequently yield correct responses. We utilize DeepSeekMath-7B-RL as the sampling model to evaluate difficulty across all experiments in the paper. Varying this sampling model to align with the generative model may further enhance performance, which we leave as future work. Notably, one of the benefits of fail rate is that it allows to reuse the sampled responses during difficulty evaluation as synthetic responses for dataset construction. See implementation details in Appendix B.2.

### 3.3 The DART-Math Datasets

We utilize DARS-Uniform and DARS-Prop2Diff to construct two datasets, DART-Math-Uniform and DART-Math-Hard respectively for instruction tuning. We use the original training queries of the GSM8K (Cobbe et al., 2021a) and MATH datasets to synthesize responses. We maintain fixed queries to better isolate the effects of difficulty-aware rejection tuning, while techniques for query augmentation, as discussed in prior studies (Yu et al., 2024), could be potentially incorporated to further improve the performance. The synthetic datasets are augmented with the original GSM8K and MATH training data to form the final datasets. We set $k_u$ in DARS-Uniform as 40 and $k_p$ in DARS-Prop2Diff as 192 to form both datasets of around 590k samples. Our data samples only involve natural language reasoning without using external tools such as code execution. Comparison of our datasets with previous datasets is illustrated in Table 1. Our datasets are generally smaller than most previous datasets, and in §4.2 we will empirically demonstrate that **the DART datasets are the most cost-effective datasets publicly available**. Remarkably, our approach solely utilizes DeepSeekMath-7B-RL to evaluate difficulty of queries and synthesize responses, without relying on ChatGPT that is commonly used in other studies.

Our approach typically requires more sampling trials than vanilla rejection sampling to generate a dataset of comparable size because difficult queries often need more samples to secure the required

number of correct responses. Despite this, it is crucial to point out that our overall training cost does not exceed that of vanilla instruction tuning. We emphasize that the data synthesis process is a one-time effort. Once the synthetic dataset is created, it can be utilized for multiple training runs across various base models. Furthermore, this dataset will be publicly available, extending its utility to a wide range of users. From this perspective, the initial higher synthesis cost is effectively amortized over numerous training runs and the broad user base, rendering the synthesis cost virtually imperceptible to individual dataset users. We will discuss the synthesis cost further in §4.3.

## 4 Experiments

### 4.1 General Setup

Below we summarize the key setup details, while we include more information in Appendix B.

**Data synthesis:** We synthesize responses using the original training queries of the MATH and GSM8K datasets. As described in §3.2, we utilize DeepSeekMath-7B-RL to synthesize all the data. We use temperature sampling with adjusted temperature to sample answer-correct responses to difficult queries. We set the maximum number of output tokens as 2048 and adopt top-p sampling with $p = 0.95$. We use chain-of-thought prompt (Wei et al., 2022) to synthesize. We use the vLLM library (Kwon et al., 2023) to accelerate the generation. In our setting, sampling 35k samples on MATH / GSM8k queries takes about 1 NVIDIA A100 GPU hour.

**Training:** We perform standard instruction tuning on our synthetic datasets `DART-Math-Uniform` and `DART-Math-Hard`, based on several base models including Llama3-8B (Meta, 2024), Mistral-7B (Jiang et al., 2023), and Llama3-70B as representatives of general models, and DeepSeekMath-7B (Shao et al., 2024) as the representative of math-specialized models. For simplicity, we keep most hyperparameters the same across different models and datasets, and tune only several key hyperparameters like learning rate and number of epochs, as detailed in Appendix B.1.

**Evaluation:** For comprehensive assessment of mathematical reasoning of the models, we adopt 6 benchmarks for both in-domain and out-of-domain (OOD) evaluation. Specifically, we use the GSM8K and MATH test set as the in-domain test. GSM8K consists of grade school arithmetic tasks and are considered much simper than MATH that contains challenging competition mathematical problems. For OOD test, we utilize the following four challenging benchmarks:

- **CollegeMath** (Tang et al., 2024): This test set contains 2818 college-level mathematical problems extracted from 9 textbooks across 7 domains such as linear algebra and differential equations, testing generalization on complex mathematical reasoning in diverse domains.

- **DeepMind-Mathematics** (Saxton et al., 2019): This test set contains 1000 problems from a diverse range of problem types based on a national school mathematics curriculum (up to age 16), testing basic mathematical reasoning in diverse domains.

- **OlympiadBench-Math** (He et al., 2024): This benchmark contains 675 Olympiad-level mathematical problems from competitions, which is a text-only English subset of Olympiad-Bench, testing generalization on the most complex mathematical reasoning.

- **TheoremQA** (Chen et al., 2023): This benchmark contains 800 problems focused on utilizing mathematical theorems to solve challenging problems in fields such as math, physics and engineering, testing generalization on theoretical reasoning in general STEM.

All results are from natural language reasoning without using external tools, through greedy decoding.

**Baselines:** We compare `DART` with the state-of-the-art instruction-tuned mathematical models such as MetaMath (Yu et al., 2024), MMIQC (Liu et al., 2024a), KPMah-Plus (Huang et al., 2024), and Xwin-Math (Li et al., 2024). We copy the results directly from the respective papers except for MetaMath and MMIQC, where we run our own training since their datasets are public. As shown in Table 1, these SOTA datasets all rely on proprietary models for data synthesis. Another ablation baseline to `DART` is vanilla rejection tuning (VRT), where we synthesize a dataset of the same size of 0.59M examples with DeepSeekMath-7B-RL, using vanilla rejection sampling as described in §2.1. We note that there are other strong models such as Yue et al. (2024); Gou et al. (2024) that are trained to solve mathematical problems utilizing code execution, we exclude them since this study focuses on reasoning without using tools.

| Model | # Samples | In-Domain | | College | Out-of-Domain | | | AVG |
|---|---|---|---|---|---|---|---|---|
| | | MATH | GSM8K | | DM | Olympiad | Theorem | |
| GPT-4-Turbo (24-04-09) | – | 73.4 | 94.5 | – | – | – | 48.4 | – |
| GPT-4 (0314) | – | 52.6 | 94.7 | 24.4 | – | – | – | – |
| Claude-3-Opus | – | 60.1 | 95.0 | – | – | – | – | – |
| Gemini 1.5 Pro | – | 67.7 | – | – | – | – | – | – |
| **70B General Base Model** | | | | | | | | |
| Llama2-70B-Xwin-Math-V1.1[†] | 1.4M | 52.5 | 90.2 | 33.1 | 58.0 | 16.3 | 14.9 | 44.2 |
| Llama3-70B-ICL | – | 44.0 | 80.1 | 33.5 | 51.7 | 10.8 | 27.0 | 41.2 |
| Llama3-70B-MetaMath | 0.40M | 44.9 | 88.0 | 31.9 | 53.2 | 11.6 | 21.9 | 41.9 |
| Llama3-70B-MMIQC | 2.3M | 49.4 | 89.3 | 37.6 | 60.4 | 15.3 | 23.5 | 45.9 |
| Llama3-70B-VRT | 0.59M | 53.1 | 90.3 | 36.8 | 62.8 | 19.3 | **28.6** | 48.5 |
| DART-Math-Llama3-70B (Uniform) | 0.59M | 54.9 ↑1.8 | **90.4** ↑0.1 | **38.5** ↑1.7 | **64.1** ↑1.3 | 19.1 ↓0.2 | 27.4 ↓1.2 | 49.1 ↑0.6 |
| DART-Math-Llama3-70B (Prop2Diff) | 0.59M | **56.1** ↑3.0 | 89.6 ↓0.7 | 37.9 ↑1.1 | **64.1** ↑1.3 | **20.0** ↑0.7 | 28.2 ↓0.4 | **49.3** ↑0.8 |
| **7B Math-Specialized Base Model** | | | | | | | | |
| DeepSeekMath-7B-ICL | – | 35.5 | 64.2 | 34.7 | 45.2 | 9.3 | 23.5 | 35.4 |
| DeepSeekMath-7B-Instruct | 0.78M | 46.9 | 82.7 | 37.1 | 52.2 | 14.2 | 28.1 | 43.5 |
| DeepSeekMath-7B-MMIQC | 2.3M | 45.3 | 79.0 | 35.3 | 52.9 | 13.0 | 23.4 | 41.5 |
| DeepSeekMath-7B-KPMath-Plus | 1.6M | 48.8 | 83.9 | – | – | – | – | – |
| DeepSeekMath-7B-VRT | 0.59M | 53.0 | **88.2** | **41.9** | 60.2 | 19.1 | 27.2 | 48.3 |
| DART-Math-DSMath-7B (Uniform) | 0.59M | 52.9 ↓0.1 | **88.2** | 40.1 ↓1.8 | 60.2 | 21.3 ↑2.2 | **32.5** ↑5.3 | 49.2 ↑0.9 |
| DART-Math-DSMath-7B (Prop2Diff) | 0.59M | **53.6** ↑0.6 | 86.8 ↓1.4 | 40.7 ↓1.2 | **61.6** ↑1.4 | **21.7** ↑2.6 | 32.2 ↑5.0 | **49.4** ↑1.1 |
| **7-8B General Base Model** | | | | | | | | |
| Llama2-7B-Xwin-Math-V1.1[†] | 1.4M | 45.5 | 84.9 | 27.6 | 43.0 | 10.5 | 15.0 | 37.8 |
| Mistral-7B-ICL | – | 16.5 | 45.9 | 17.9 | 23.5 | 3.7 | 14.2 | 20.3 |
| Mistral-7B-WizardMath-V1.1 (RL) | – | 32.3 | 80.4 | 23.1 | 38.4 | 7.7 | 16.6 | 33.1 |
| Mistral-7B-MetaMath | 0.40M | 29.8 | 76.5 | 19.3 | 28.0 | 5.9 | 14.0 | 28.9 |
| Mistral-7B-MMIQC | 2.3M | 37.4 | 75.4 | 28.5 | 38.0 | 9.4 | 16.2 | 34.2 |
| Mistral-7B-MathScale | 2.0M | 35.2 | 74.8 | 21.8 | – | – | – | – |
| Mistral-7B-KPMath-Plus | 1.6M | **46.8** | 82.1 | – | – | – | – | – |
| Mistral-7B-VRT | 0.59M | 38.7 | 82.3 | 24.2 | 35.6 | 8.7 | 16.2 | 34.3 |
| DART-Math-Mistral-7B (Uniform) | 0.59M | 43.5 ↑4.8 | **82.6** ↑0.3 | 26.9 ↑2.7 | 42.0 ↑6.4 | 13.2 ↑4.5 | 16.4 ↑0.2 | 37.4 ↑3.1 |
| DART-Math-Mistral-7B (Prop2Diff) | 0.59M | 45.5 ↑6.8 | 81.1 ↓1.2 | **29.4** ↑5.2 | **45.1** ↑9.5 | **14.7** ↑6.0 | **17.0** ↑0.8 | **38.8** ↑4.5 |
| Llama3-8B-ICL | – | 21.2 | 51.0 | 19.9 | 27.4 | 4.2 | 19.8 | 23.9 |
| Llama3-8B-MetaMath | 0.40M | 32.5 | 77.3 | 20.6 | 35.0 | 5.5 | 13.8 | 30.8 |
| Llama3-8B-MMIQC | 2.3M | 39.5 | 77.6 | **29.5** | 41.0 | 9.6 | 16.2 | 35.6 |
| Llama3-8B-VRT | 0.59M | 39.7 | 81.7 | 23.9 | 41.7 | 9.3 | 14.9 | 35.2 |
| DART-Math-Llama3-8B (Uniform) | 0.59M | 45.3 ↑5.6 | **82.5** ↑0.8 | 27.1 ↑3.2 | **48.2** ↑6.5 | 13.6 ↑4.3 | 15.4 ↑0.5 | 38.7 ↑3.5 |
| DART-Math-Llama3-8B (Prop2Diff) | 0.59M | **46.6** ↑6.9 | 81.1 ↓0.6 | 28.8 ↑4.9 | 48.0 ↑6.3 | **14.5** ↑5.2 | **19.4** ↑4.5 | **39.7** ↑4.5 |

Table 2: Main results on mathematical benchmarks. College, DM, Olympiad, Theorem denote the CollegeMath, DeepMind-Mathematics, OlympiadBench-Math, TheoremQA benchmarks respectively. We annotate the absolute accuracy change compared to the VRT baseline within the same base model. Bold means the best score within the respective base model. ICL, MetaMath, MMIQC, and VRT baselines are from our own runs, while other numbers are copied from the respective papers or reports. For WizardMath and Xwin-Math, we take the public model checkpoints and evaluate ourselves using their official CoT prompt. [†]: For Xwin-Math, we take the best public models that are based on Llama2 (Touvron et al., 2023), which is not a very fair comparison with others.

## 4.2 Main Results

**Comparing with Vanilla Rejection Tuning:** The main results are in Table 2. DART-Math based on all four different base models outperforms the VRT baselines on most benchmarks consistently. Focusing on performance with 7-8B general base models, DART-Math-Llama3-8B (Uniform) surpasses the VRT baseline across all 6 benchmarks by an average of 3.5 absolute points, while DART-Math-Llama3-8B (Prop2Diff) achieves an average improvement of 4.5 points. On the in-domain challenging MATH benchmark, DART-Math (Prop2Diff) enhances performance over VRT by nearly 7 absolute points for both Mistral-7B and Llama3-8B models. For OOD benchmarks, DART-Math (Prop2Diff) shows particularly notable gains on more difficult benchmarks, with improvements ranging from 5.2 to 9.5 absolute points on CollegeMath, DeepMind-Mathematics, and OlympiadBench-Math. This indicates effective generalization of our approach. These improvements over the VRT baselines demonstrate the effectiveness of the proposed difficulty-aware rejection sampling. We note that DART-Math does not greatly boost the relatively simple, in-domain GSM8K benchmark. This is expected, as explained in §2.2, because vanilla rejection tuning expected does not face severe bias issues like those seen in more challenging datasets. Thus, difficulty-aware rejection sampling has a limited impact on easy datasets. Interestingly, on much stronger base models DeepSeekMath-7B and Llama3-70B, the improvement margin of DART-Math over VRT narrows, with about a 1-point gain on average. We hypothesize that this is due to these models' extensive pretraining on mathematical content. This pretraining likely covers most skills that could be learned from the GSM8K and MATH training queries, suggesting that the query set itself, rather than the

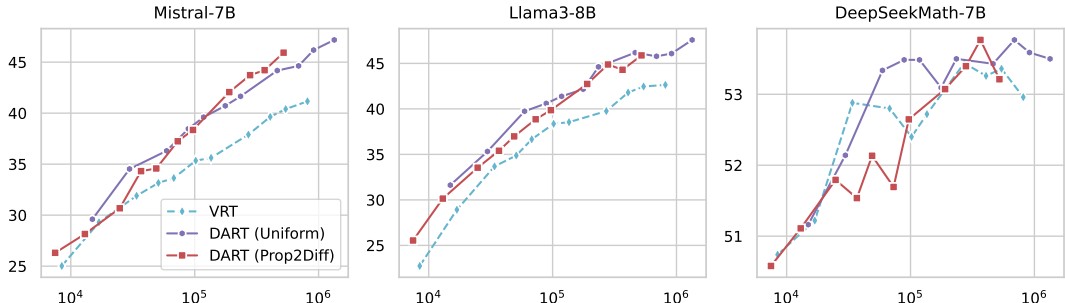

Figure 3: Scaling curves of MATH test performance against number of training samples synthesized from MATH training queries, training is on three base models.

responses, becomes the bottleneck. Thus augmenting the range of queries could be a more effective strategy for future improvements.

**Comparison with previous top-performing methods:** `DART-Math` achieves superior or comparable performance to previous best models. Specifically, when compared with MetaMath, `DART-Math` wins greatly in all cases. Additionally, `DART-Math-DSMath-7B` achieves the state-of-the-art results for models sized 7-8B on challenging benchmarks such as MATH, OlympiadBench-Math, and TheoremQA. On average, `DART-Math-Mistral-7B` (Prop2Diff) surpasses Mistral-7B-MMIQC by 4.6 absolute points, despite using only a quarter of its training sample size. Compared with concurrent work KPMath-Plus which relies on GPT-4 and has not released either the data or the model, our approach slightly underperforms on Mistral-7B for GSM8K and MATH. However, `DART-Math` excels against it on DeepSeekMath-7B by a significant margin, utilizing around one-third of its training data size. The Xwin-Math models perform well on the GSM8K benchmark but fall behind `DART-Math` (Prop2Diff) on other challenging benchmarks overall, particularly with a more pronounced gap on 70B models — although we note that their models are based on Llama2 which is not very fair to compare with. Importantly, we fully open-source our datasets and models, designating both `DART-Math-Uniform` and `DART-Math-Hard` as the **best-performing and most cost-effective public instruction tuning datasets available for advancing mathematical problem-solving.**

**Additional results:** For additional results, such as domain-wise performance on MATH and comparison to RL, we refer readers to Appendix C.

## 4.3 Analysis

**Scaling behaviors of different data synthesis methods:** We study the scaling behaviors of our data synthesis approach and compare it to vanilla rejection sampling. As described in 2.2, our method is motivated to mitigate the bias towards easy queries that are only pronounced in challenging datasets. Therefore, in the scaling experiment we only synthesize responses for the training queries of the challenging MATH dataset and report the performance on the MATH test set. Figure 3 presents the results across three different base models as we scale the training data size from thousands to nearly 1 million samples. We observe a steady improvement in performance as the training data size increases exponentially. `DART` consistently outperforms VRT on general base models Mistral-7B and Lllama3-8B, achieving better scaling. On DeepSeekMath-7B, however, the performance differences between various approaches are minimal. Observing the absolute accuracy changes, DeepSeekMath-7B already achieves over 50% accuracy with just thousands of training samples, and scaling up to 1 million samples leads to only a modest 3-point improvement. This is in stark contrast to the over 20-point improvements seen on other models like Mistral-7B and Llama3-8B. As discussed in §4.2, we believe this phenomenon is due to the MATH training queries not being particularly beneficial for DeepSeekMath-7B, which has undergone extensive math-specific continual pretraining. Consequently, for DeepSeekMath-7B, the differences between these approaches are not significant, and the main bottleneck shifts to query coverage rather than the responses themselves.

**Effect of one-response coverage:** In §3.2, we describe that `DARS-Prop2Diff` can cause zero synthetic responses for easy queries, especially when the number of training samples is small. Therefore, we ensure that the easy queries have at least one correct response practically. Here we examine the impact of this one-response coverage by comparing the Prop2Diff strategy with and

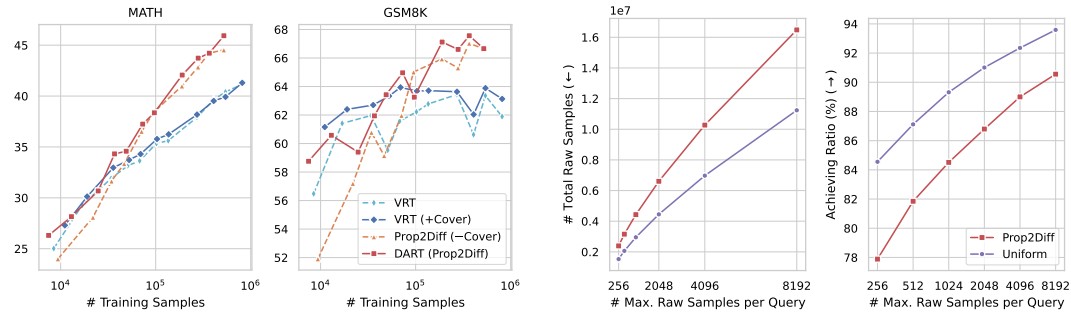

Figure 4: **From Left to Right, (1) and (2):** Scaling curves studying the effect of one-response coverage. "Prop2Diff (−Cover)" denotes `DARS-Prop2Diff` without enforcing at least one synthetic response for each query, while "VRT (+Cover)" denotes vanilla rejection sampling enforcing at least one synthetic response for each query. **(3) and (4):** The total number of raw samples needed, and the actual ratio ($r$) of queries achieving the desiderata of the two `DARS` synthesis strategy for 585k-sized dataset curation respectively, when we vary the maximum allowable raw samples per query ($n_{max}$).

without this coverage constraint, as training data sizes increase. Figure 4 (Left) displays the outcomes on the MATH and GSM8K benchmarks respectively. As anticipated, when the training data size is relatively small, the one-response coverage proves beneficial, particularly on the simpler GSM8K benchmark, improving accuracy by about 8 points. This suggests that effective learning for easy problem-solving can be achieved with just one additional correct response. As we scale up the training data size, the natural increase in coverage for easy queries causes that the difference between the two approaches diminishes. Additionally, we explore the implementation of one-response coverage in vanilla rejection tuning to determine if adding one synthetic response for difficult queries could address its issue of low coverage for such queries. However, this modification does not significantly aid in learning difficult queries, as observed on the challenging MATH benchmark. This indicates that complex problems generally require a greater number of training samples for effective learning.

**Synthesis cost:** `DART` generally needs more sampling trials to synthesize the same size of dataset compared to vanilla rejection tuning, as discussed in §3.3. It is important to underline that the synthesis cost, although initially higher, is a one-time expense. Once the dataset is synthesized, it can be used by the community and us to train numerous models, effectively amortizing the cost. To provide a quantitative understanding of the synthesis cost, we consider two main factors: $n_{max}$, the maximum allowable raw samples for each query, and $r$, the ratio of queries that achieve the designated number of responses. If $n_{max}$ is set too high, sampling may continue indefinitely for particularly difficult or noisy queries, resulting in a high synthesis cost. Conversely, a too small $n_{max}$ may result in many queries not gathering the sufficient number of correct responses, leading to a lower $r$. Figure 4 (Right) illustrates the total number of raw samples required to synthesize 585k examples and the query achieving ratio $r$ as we increase $n_{max}$. When $n_{max}$ reaches 2048, over 90% of the queries can collect the designated number of responses under `DARS-Uniform`, with a corresponding total number of samples around 5 million. To reach 90% achieving ratio for `DARS-Prop2Diff`, $n_{max}$ needs to be at least 8K, and the total number of raw samples exceeds 15 million. In our experiments, we achieved an over 95% ratio $r$, sampling approximately 150 million samples in total, which required running inference of DeepSeekMath-7B-RL for about 160 NVIDIA A100 GPU days. Besides that synthesis is a one-time cost, we would like to emphasize the number of samples is not a fair metric to compare synthesis cost between different works — our synthesis model of 7B size is relatively inexpensive and fast to run, compared to the much more costly and slower GPT-4 used in most previous studies. Moreover, achieving a query ratio as high as 95% may not be necessary to reach good performance. A slightly lower ratio of 85% or 90% might not significantly impact performance but could substantially reduce the synthesis cost. We plan to explore this balance further in future work.

## 5 Discussion

In this paper, we focus on instruction tuning for mathematical problem solving, and discuss the impact of distribution and coverage of training queries across different difficulties. We identify the bias towards easy queries in vanilla rejection tuning, and propose difficulty-aware rejection tuning, `DART`, as a remedy. Based on our approach, we create and open-source the best-performing and

the most cost-effective instruction tuning datasets for mathematical reasoning, without relying on proprietary models. Extensive experiments across various base models and benchmarks demonstrate the effectiveness of our approach.

**Limitations:** We utilize fail rate as the difficulty metric, yet it may be sub-optimal. Other metrics such as direct scoring (Liu et al., 2024b), Elo ratings, or the minimum pretraining compute to train a model that can always answer correctly (Burns et al., 2023) may be further explored. `DART-Math` is limited by natural language reasoning, while it is shown that generating and executing code helps solve mathematical problems significantly (Zhou et al., 2024; Yue et al., 2024; Gou et al., 2024; Liao et al., 2024; Toshniwal et al., 2024) — we think the bias in vanilla rejection sampling also exists for code generation, and `DART` could be integrated to potentially improve code generation as well.

## Acknowledgments

We thank Zhiyuan Zeng, Wei Xiong and Chenyang Zhao for helpful discussions. Yuxuan is partially supported by Tsinghua University Initiative Scientific Research Program (Student Academic Research Advancement Program).

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

# A    Comparison to Methods Based on Non-vanilla Rejection Sampling

Though both ToRA and MARIO have not released their datasets and focus on mathematical problem-solving using code in addition to natural language, which is out of our scope and thus not comparable, we try to implement the natural-language versions of their data synthesis strategies, which are comparable with `DARS`.

## A.1    `DARS` produces distributions not biased towards easy queries

The most important difference between `DARS` and ToRA/MARIO is how responses are distributed across various queries — while we adjust the distribution either to be uniform or to favor more difficult queries, rather than merely improving coverage, **ToRA/MARIO mainly focus on improving coverage without managing the distribution explicitly, leading to datasets that may still bias towards easy queries**.

As shown in Table 3, though the absolute numbers of responses are not directly comparable between different methods, distribution-wise we can see that ToRA/MARIO still produce fewer responses for difficult problems than the easy ones. This especially contrasts with `DART-Math-Hard`, which produces, for example, 10x more responses for the MATH Level 5 queries than for the GSM8K queries.

As demonstrated in Figure 4 (Left), a high coverage rate (VRT+Cover) alone does not guarantee superior performance.

| Synthetic Dataset | Size | RPQ in | RPQ in Level-wise MATH | | | | | MATH |
|---|---|---|---|---|---|---|---|---|
| | (k) | GSM8K | 1 | 2 | 3 | 4 | 5 | Coverage |
| ToRA | 72 | 5.03 | 5.01 | 4.99 | 4.95 | 4.77 | 3.84 | 93.4% |
| MARIO | 29 | 2.02 | 2.01 | 1.98 | 1.94 | 1.89 | 1.57 | 91.3% |
| DART-Math-Uniform | 585 | 39.93 | 40.00 | 40.00 | 39.80 | 39.54 | 37.14 | 99.6% |
| DART-Math-Hard | 590 | 8.49 | 14.28 | 33.52 | 54.94 | 79.59 | 107.06 | 99.6% |

Table 3: Comparison between datasets synthesized by methods based on non-vanilla rejection sampling. "RPQ" means the average number of responses per query. The ToRA and MARIO datasets here are implemented by us according to their papers' descriptions, since the official implementations have not been open-sourced.

## A.2    `DARS` achieves high coverage even on the hardest queries

It is worth noting that a relatively high total coverage on MATH training set does not mean that the hard queries are well covered. For example, the MetaMathQA-MATH-AnsAug dataset achieves 82.8% of coverage on the MATH training set with evenly allocated budgets yet still admits missing a significant portion of hard queries and biasing towards easy queries, as analyzed in Figure 2.

In Table 4 we show the coverage rate across all the difficulty levels by different methods. The ToRA-Corpus-16k statistics show that it only covers 68% of the Level 5 MATH queries while DART-Math datasets cover 99.6%.

## A.3    Details of Re-implementing Data Synthesis Strategies of ToRA and MARIO

Here we supplement more details on how we replicate the ToRA/MARIO synthesis pipeline to conduct the analysis present in the general author rebuttal. Below we show in the format as "ToRA/MARIO's method -> how we adapt similar spirits for a simpler replication" step by step (we use CoT format rather than tool-integrated reasoning for a fairer comparison with our datasets):

**ToRA:**

---

[3]`https://openreview.net/forum?id=Ep0TtjVoap`

| MATH training set coverage | Total | Level 1 | Level 2 | Level 3 | Level 4 | Level 5 |
|---|---|---|---|---|---|---|
| ToRA-Corpus-16k-MATH | 83.1% | 97.7% | 91.6% | 86.5% | 81.3% | 68.0% |
| MetaMath-MATH-AnsAug | 82.8% | 98.1% | 93.6% | 86.7% | 76.6% | 48.9% |
| VRT Baseline | 84.9% | 99.6% | 98.2% | 95.2% | 89.8% | 62.9% |
| DART-Math-* | **99.6%** | **100.0%** | **100.0%** | **99.9%** | **99.7%** | **99.1%** |

Table 4: MATH training set coverage rates across all the difficulty levels of different synthetic datasets. The numbers of ToRA-Corpus-16k-MATH are from their OpenReview page[3]. The two DART-Math-* datasets have the same coverage because of the "Cover" operation, which tries to ensure there is at least one correct response for each query.

1. Once for each problem in MATH&GSM8K with GPT-4, keeping the correct responses. -> We follow this with GPT-4o mini[4]

2. 10 trials for each problem not correctly answered by greedy decoding with GPT-4 and keeping up to 4 correct responses per problem (to form ToRA-Corpus-16k). -> We follow this with GPT-4o mini.

3. Training CodeLlama models on ToRA-Corpus-16k to perform rejection sampling next. -> To avoid additional training for a fairer comparison, we use DeepSeekMath-7B-RL to replace the trained CodeLLama models here to align with DART-Math.

   (a) 64 trials for each problem in MATH&GSK8K with CodeLlama, getting 233k distinct correct responses. -> We follow this with DeepSeekMath-7B-RL, getting 733k distinct correct responses.

   (b) Correcting wrong responses by greedy decoding from the correct preceding portions (costing no more than 64 trials for each problem) with CodeLLaMA-34B, getting 69k corrected responses. -> We simplify this by re-sampling another up to 64 trials per problem for all the incorrect responses, getting 225k correct samples.

   (c) Both ToRA and our adaptation: Randomly selecting up to 4 correct responses per problem from steps (a) and (b).

4. Merge ToRA-Corpus-16k and data from step 3 to form the final training dataset of 69k responses. -> We exactly follow this to form the final dataset of 72k responses.

**MARIO:**

1. Greedy decoding using GPT3.5 and GPT-4 each once for MATH&GSM8K, getting two responses for each query, only correct ones are kept -> We follow this but use GPT-4o mini to sample two responses for each query.

2. Sampling for 2 trials for each problem not correctly answered in step 1 using GPT-4, only correct ones are kept -> We follow this with GPT-4o mini.

3. Manually correcting responses for part of the remaining problems, then tuning Llemma-34B on it to obtain a synthesis agent for next steps -> this involves human annotation and is not comparable to our approach. For simplicity, we adopt DeepSeekMath-7B-RL as the synthesis agent to align with the DART-Math datasets.

4. Sampling with 100 trials and keeping up to 4 correct responses per problem for the remaining unanswered MATH queries, achieving 93.8% coverage on MATH -> we follow this and achieve 91.3% coverage on MATH.

5. Sampling with 1 trial for new problems introduced by MetaMath and keeping correct ones -> this step introduces new prompts and would only skew the distribution of responses, if any, towards easy queries. We remove this step for simplicity, which would not affect our conclusion.

---

[4]For GPT-4o mini, we use the version of `gpt-4o-mini-2024-07-18` by default.

# B  Experimental Setup

## B.1  Training Setup

We train all the models using the Transformers library (Wolf et al., 2019).

**Sequence Packing:**  To efficiently save computation wasted by padding tokens, we employ sequence packing (Krell et al., 2021). We shuffle all samples in each epoch before sequence packing, ensuring that the same semantic sequences are not always in the same computation sequence.

**Batch Size:**  The computation sequence token length is set to 4096, considering that most sequences in the training datasets are shorter than this length. The batch size is 64, though there are usually more than 64 samples in one batch because one computation sequence can pack multiple semantic sequences. We disable gradient accumulation (Lin et al., 2018) by default, but when the memory is not sufficient, we increase the number of gradient accumulation steps and keep other settings unchanged. Specifically, we use 2 gradient accumulation steps when training Llama3-8B on 8 NVIDIA A100 GPUs under our setting.

**Learning Rate:**  We use the Adam optimizer (Zhang, 2018) with the weight decay as 0. We use a linear warmup with a warmup step ratio of 0.03 and cosine learning rate scheduler. The maximum learning rates are set as follows: Mistral-7B at 1e-5, DeepSeekMath-7B and Llama3-8B at 5e-5, and Llama3-70B at 2e-5. We determine the values by searching through `1e-6,5e-6,1e-5,2e-5,5e-5,1e-4` according to the MATH performance after training on MMIQC for 1 epoch.

**# Training Epochs:**  The default number of epochs is 3. For MMIQC, we train for 1 epoch following Liu et al. (2024a). For Llama3 models, we train for 1 epoch because preliminary experiments indicate that 1 epoch consistently outperforms 3 epochs.

**Prompt Template:**  For the prompt template, we use the format following Taori et al. (2023):

> **Prompt Template**
>
> ```
> Below is an instruction that describes a task.  Write a response that
> appropriately completes the request.\n\n###Instruction:\n{query}\n\n###
> Response:\n
> ```

**Other Details:**  For efficiency, We utilize various tools / libraries / techniques including:

- the DeepSpeed distributed framework (Rasley et al., 2020) with ZeRO (Rajbhandari et al., 2020) stage 3
- gradient checkpointing (Chen et al., 2016)
- `torch.compile` (Ansel et al., 2024)
- mixed-precision training (Micikevicius et al., 2018) of BrainFloat16 (Kalamkar et al., 2019) and TensorFloat32 (NVIDIA, 2020)

**Hardware:**  For 7B or 8B models, we train on 8 NVIDIA A100 GPUs. For 70B models, we train on 32 NVIDIA A100 GPUs.

**Training Time Cost**  The specific training time cost depends on too many factors to give a precise expression, such as model architecture, model size, data content, training algorithm implementation, hardware environment, etc. Here we provide several data points under our setting for reference:

## B.2  Synthesis Setup

**Generation:**  We utilize the vLLM library Kwon et al. (2023), setting the maximum number of output tokens as 2048 and adopt top-p sampling with $p = 0.95$. For temperature $t$, we search from 0.3 to 1.8 with a step of 0.1 by using DeepSeekMath-7B-RL to sample answer-correct responses to queries in MATH training set. We observe the speeds to achieve specified correct answer coverage of different temperatures and find that, for DeepSeekMath-7B-RL, higher temperatures achieve faster, but $t \geq 1.0$ are quite similar and $t \geq 1.7$ cause the output to be nonsense. Besides, we find that higher temperatures produce more diverse responses by visualizing the embedings of response from

| Dataset | # Samples (k) | Model | Hardware | Time (hour/epoch) |
|---------|---------------|-------|----------|-------------------|
| `DART-Math-Hard` | 585 | DeepSeekMath-7B | 8 A100 GPUs | 3 |
| `DART-Math-Hard` | 585 | Mistral-7B | 8 A100 GPUs | 3 |
| `DART-Math-Hard` | 585 | Llama3-8B | 8 A100 GPUs | 3 |
| `DART-Math-Hard` | 585 | Llama3-70B | 32 A100 GPUs | 6 |

Table 5: Examples of training time cost.

different temperatures to the same query using t-SNE (Van der Maaten & Hinton, 2008). Finally, we set the temperature as $t = 1.6$. This choice should be fair since the temperature search is not specifically tailored for DART.

**Grading:** To judge whether the answers in raw responses are correct or not as accurately as possible, we implement an elaborate answer extraction and judgement pipeline based on regular expressions and SymPy (Meurer et al., 2017) symbolic calculation, which is able to correctly process most mathematical objects such as matrices (vectors), intervals, symbols besides numbers, as well as some special texts like bool expressions, dates and times.

**Calculating Fail Rate:** For efficiency, we merge `DARS-Uniform` synthesis and calculating fail rates as mentioned in §3.2. Specifically, we set $k_u = 192$ to synthesize our data pool, and based on all the responses sampled, we calculate fail rate for each query as

$$\text{fail rate} = \frac{\text{\# all correct responses}}{\text{\# all raw responses}}$$

which would produce more accurate fail rate values but is not necessary for general algorithm implementations.

### B.3 Evaluation Setup

**Generation** Like §B.2, we use the vLLM library, setting the maximum number of output tokens as 2048 and adopting top-p sampling with $p = 0.95$. But we use greedy decoding (i.e. set temperature $t = 0$) for evaluation. Note that there might still be randomness from vLLM implementation despite using greedy decoding, so we run each evaluation in §2 with at least 3 random seeds. When evaluating models trained by us, we use the Alpaca (Taori et al., 2023) prompt template consistent with training as shown in §B.1. All SFT & RL models are evaluated with 0-shot, while all base models with few-shot in-context learning (ICL): MATH (4-shot), GSM8K (4-shot), CollegeMath (4-shot), DeepMind Mathematics (4-shot), OlympiadBench-Math (4-shot), TheoremQA (5-shot). For baseline models, prompts in official implementations are used. Specially, the CoT version of Alpaca prompt template is used for WizardMath.

**Grading** We utilize the same pipeline as §B.2 by default, except that, for OlympiadBench, we use the official implementation of answer correctness judgement component by He et al. (2024), which utilizing the numerical error range information provided with query, but keep the answer extraction component of ours, because the official implementation fails to extract a non-negligible part of answers, especially for base model ICL.

## C Additional Results

### C.1 Domain-wise Performance on MATH

We test the domain-wise performance on MATH for rejection-tuned models based on Mistral-7B and Llama3-8B. As shown in Table 6, both domain-wise and domain-macro-average scores still show DART's significant improvement across all domains.

| Model | MATH Domains | | | | | | | Average | |
|---|---|---|---|---|---|---|---|---|---|
| | Prob. | Prealg. | Num. | Interm. Alg. | Alg. | Precalc. | Geo. | Micro | Macro |
| Llama3-8B-VRT | 34.2 | 57.8 | 30.7 | 20.4 | 59.6 | 22.5 | 29.0 | 39.7 | 36.3 |
| DART-Math-Llama3-8B (Uniform) | 34.6 | **65.7** | 35.7 | 25.4 | 66.6 | 29.3 | 32.4 | 45.3 | 41.4 |
| DART-Math-Llama3-8B (Prop2Diff) | **38.8** | 62.9 | **36.8** | **26.1** | **67.3** | **32.0** | **39.9** | **46.6** | **43.4** |
| Mistral-7B-VRT | 32.1 | 56.3 | 29.6 | 19.0 | 58.4 | 22.2 | 30.7 | 38.7 | 35.5 |
| DART-Math-Mistral-7B (Uniform) | 33.8 | 59.8 | 35.2 | 24.4 | 64.1 | 28.8 | 34.2 | 43.5 | 40.0 |
| DART-Math-Mistral-7B (Prop2Diff) | **36.1** | **61.3** | **35.4** | **26.0** | **65.7** | **31.1** | **40.5** | **45.5** | **42.3** |

Table 6: MATH performance across all the domains. Macro average assigns equal weights to each domain, while micro average assigns equal weights to each query, which is the same to the whole-benchmark score. The full names of the domains are Counting & Probability, Prealgebra, Number Theory, Intermediate Algebra, Algebra, Precalculus, Geometry, respectively. **Bold** means the best score within the respective base model.

## C.2 DART achieves comparable performance with RL

DART is an SFT method, which is usually not comparable with RL method like GRPO used by DeepSeekMath-7B-RL.

However, even considering comparison with DeepSeekMath-7B-RL, we find that sole SFT with DART can produce performance comparable with RL on DeepSeekMath-7B, as shown by Table 7.

| Model | MATH | GSM8K | College | DM | Olympiad | Theorem | AVG |
|---|---|---|---|---|---|---|---|
| DeepSeekMath-7B-RL | 53.1 | **88.4** | 41.3 | 58.3 | 18.7 | **35.9** | 49.3 |
| DART-Math-DSMath-7B (Uniform) | 52.9 | 88.2 | 40.1 | 60.2 | 21.3 | 32.5 | 49.2 |
| DART-Math-DSMath-7B (Prop2Diff) | **53.6** | 86.8 | 40.7 | **61.6** | **21.7** | 32.2 | **49.4** |

Table 7: Performance by DART and RL on DeepSeekMath-7B. College, DM, Olympiad, Theorem denote the CollegeMath, DeepMind-Mathematics, OlympiadBench-Math, TheoremQA benchmarks respectively. **Bold** means the best score within the respective base model.

# D  Related Work

**Rejection-Sampling-Based Data Synthesis:**  Rejection sampling (Neal, 2003) is a statistical approach used to generate samples from some target distribution that is not directly accessible (e.g., the distribution of correct responses to all the queries). In model training, this can be used for constructing training data and usually implemented in some form of "sampling and filtering". Depending on the task, the supervision signal for filtering can be reward models, ground-truth answers, answer consistency, e.t.c. (Bai et al., 2022; Zelikman et al., 2022; Huang et al., 2022; Dong et al., 2023; Gulcehre et al., 2023; Yuan et al., 2023; Singh et al., 2023). However, most of previous works sample the same number of candidates for each query, regardless of the query difficulty, unconsciously introducing a bias towards easy queries in the final training data distribution. DART resolves this issue by explicitly controlling the final distribution with adaptive budget allocation of candidate samples.

**Data Construction for Instruction Tuning**  Data have been seen one of the most critical factor for the performance of instruction tuning. Previous works construct metrics for data selection and construction in diverse ways, such as training predictors (Cao et al., 2023; Lu et al., 2024; Liu et al., 2024b), prompting LLMs (Chen et al., 2024), gradient-based metrics (Xia et al., 2024) and heuristics (Li et al., 2023a,b; Ning et al., 2024). But most of them do not consider the final distribution of training data. DART focus on the metric for difficulty and further controls the whole distribution, providing a new perspective for data selection and construction.

