# OpenReview forum: "DART-Math: Difficulty-Aware Rejection Tuning for Mathematical Problem-Solving"
_NeurIPS.cc/2024/Conference — NeurIPS 2024 poster_

### Official Review · Reviewer_quxz · 2024-07-08

**Soundness:** 2
**Presentation:** 3
**Contribution:** 1
**Rating:** 4
**Confidence:** 5

**Summary:**

This paper synthesizes a math reasoning dataset with a designed way of rejection sampling. Many base models show performance improvements on math reasoning tasks after instruction-tuning on this dataset. They promise to release the dataset and models.

**Strengths:**

Their curated dataset achieves relatively good instructing-tuning performance with least data amount compared to other baselines. The dataset will be released.

**Weaknesses:**

1. The proposed sampling technique is trivial and incremental, when comparing with previous works, e.g., the uniform method is used in ToRA, and the prop2diff method is used in MARIO.
2. There’s little improvement or even performance drop when tuning Mistral-7B and DeepSeekMath-7B
compared to other baselines. As mentioned in the analysis section, this dataset is somehow replaceable by math-specific continual pre-training + supervised fine-tuning (SFT).
3. The major concern is that even the paper claims the proposed dataset is smaller, however, the LLM used to synthesize the smaller dataset is `DeepSeekMath-7B-RL`, which is trained on a larger SFT dataset. An alternative and reasonable response generation method should be leveraging the `DeepSeekMath-7B-Base` with proper prompting, as `DeepSeekMath-7B-Base` has not been supervised fine-tuned.

**Questions:**

1. What’s the query coverage ratio on MATH training set constructed by Prop2Diff?
2. Any figure or statistics to show the difficulty distribution of your DART dataset?
3. Any case studies to show the generated responses of your DART dataset? How do you extract answers in the raw response? Responded texts from the LLM are quite likely not to follow your instruction as you apply such a high temperature in sampling process. It’s not likely that simple pipelines, such as regular expressions can achieve this.

---

> ### Author Rebuttal · Authors · 2024-08-07
>
> Thanks for your comments! We address your concerns below.
>
> > **Q1**: The proposed sampling technique is trivial and incremental …… the uniform method is used in ToRA, and the prop2diff method is used in MARIO.
>
> **A1**: We respectfully disagree with the reviewer. Both ToRA and MARIO are distinct from our approach on how responses are distributed across various queries. While we adjust the distribution either to be uniform or to favor more difficult queries, ToRA&MARIO focus mainly on improving coverage without managing the distribution explicitly, leading to datasets that may still bias towards easier queries as we show quantitatively in the general author rebuttal. Please check the general author rebuttal for a summary of our point on this concern.
>
> Here we supplement more details on how we replicate the MARIO synthesis pipeline to conduct the analysis present in the general author rebuttal (the ToRA synthesis pipeline is described in response A2 to Reviewer neCQ). Below we describe in the format as “how MARIO synthesizes data -> how we replicate a simplified version of it” step by step:
>
> ---
> ---
>
> 1. Greedy decoding using GPT3.5 and GPT-4 each once for MATH&GSM8K, getting two responses for each query, only correct ones are kept -> We follow this but use GPT-4o mini to sample two responses for each query.
> 2. Sampling for 2 trials for each problem not correctly answered in 1. using GPT-4, only correct ones are kept -> We follow this with GPT-4o mini.
> 3. Manually correcting responses for part of the remaining problems, then tuning Llemma-34B on it to obtain a synthesis agent for next steps ->  this involves human annotation and is not comparable to our approach.  For simplicity, we adopt DeepSeekMath-7B-RL as the synthesis agent to align with DART-Math.
> 4. Sampling with 100 trials and keeping up to 4 correct responses per problem for the remaining unanswered MATH queries, achieving 93.8% coverage on MATH -> we follow this and achieve 91.3% coverage on MATH.
> 5. Sampling with 1 trial for new problems introduced by MetaMath and keeping correct ones -> this step introduces new prompts and would only skew the distribution of responses, if any, towards easy queries. We remove this step for simplicity, which would not affect our conclusion.
>
> ---
> ---
>
> While MARIO performs more sampling trials for difficult problems, it mainly uses coverage rate as the criterion, and a high coverage rate can be obtained while the distribution is still biased towards easy queries.
>
> We show the average # of responses per problem for different difficulty levels in the table of the general rebuttal. Our replicated ToRA and MARIO datasets obtain a similar MATH coverage ratio and total size to their original ones — While the absolute # of responses is not directly comparable between different methods, distribution-wise we can see that ToRA/MARIO still produce fewer responses for difficult problems than the easy ones. **This contrasts sharply with `DART-MATH-Hard`, which produces, for example, 10x more responses for the MATH Level 5 queries than for the GSM8K queries.**
>
> > **Q2**: There’s little improvement or even performance drop when tuning DeepSeekMath-7B
>
> **A2**: While an improvement over the VRT baseline is smaller for Llama3-70B and DeepSeekMath-7B, we note that our results are already much better than the ones from the best open-source datasets (around 4-5 points better on average) on all 4 models. Thus the release of our datasets themselves is a significant contribution to the open-source community -- **`DART-Math` is the SotA, public CoT datasets for math problem solving and much better than existing datasets across all the assessed models.**
>
> > **Q3**: this dataset is somehow replaceable by math-specific continual pre-training + SFT
>
> **A3**: This may be true, but we don't think it is a major issue. Strictly speaking, we suspect most carefully curated math SFT datasets are somehow replaceable by extensive continual pretraining + simpler SFT, yet this fact does not diminish the significance of these SFT datasets -- Continual pretraining typically occurs on a much larger scale, for instance, DeepSeekMath involves continual pretraining on 150B math-specific tokens, whereas `DART-Math` contains only \~0.2-0.3B tokens. Given this scale, the efficiency provided by SFT datasets remains crucial and non-replaceable for most developers.
>
> > **Q4**: even the paper claims the proposed dataset is smaller, however, the LLM used to synthesize the smaller dataset is `DeepSeekMath-7B-RL`, which is trained on a larger SFT dataset
>
> **A4**: We don’t think using an instructed model is a major issue as we aim to create the best synthetic SFT dataset. This setting aligns with most related works that use GPT-4, an aligned model. When we claim the proposed dataset is smaller, we suggest the reviewer note that our baselines all use GPT-4/3.5 to generate data; these models are hypothetically trained on an even larger SFT&RL dataset than DeepSeekMath-7B-RL.
>
> > **Q5**: What’s the query coverage ratio on MATH training set constructed by Prop2Diff?
>
> **A5**: 99.6%
>
> > **Q6**: Any figure or statistics to show the difficulty distribution of your DART dataset?
>
> **A6**: Figure 1 (Right).
>
> > **Q7**: Any case studies to show the generated responses of your DART dataset? How do you extract answers in the raw response?
>
> **A7**: The answers are mostly extractable after we improve existing regular expression implementation to be more comprehensive. Please refer to Table 1 in the PDF of the general rebuttal for several cases of responses for the hardest MATH problems.

---

> > ### Comment · Reviewer_quxz · 2024-08-07
> >
> > Thanks for the feedback from the authors.
> >
> > I'm not arguing that your data generated by DeepSeek-RL should be compared with previous works with GPT generated data. I think such self knowledge distillation (KD) method is fine.
> >
> > My main concern is that you use DeepSeek-RL to create the SFT dataset, then you fine-tune DeepSeek-base on this dataset, but the performance in Table 2 indicates that the the SFT model is comparable or even worse than the original DeepSeek-RL (GSM8K 86.7, MATH 58.8). So why not directly use the open-sourced DeepSeek-RL?
> >
> > The expectation of self-KD method is to significantly improve the model itself. If the self-KD cannot improve the model itself, there should be some technical or practical flaws in the method.

---

> > > ### Author Response · Authors · 2024-08-12
> > > **Urgent Reminder of Rebuttal**
> > >
> > > Dear Reviewer quxz,
> > >
> > > Sorry to disturb you for the last time, but only one day is left until the end of the reviewer-author discussion stage. We still do not know if you have received our newest response. To address your concerns, we wrote all the responses in detail and added new experiments to support them, including:
> > >
> > > 1. **A1 & Author Rebuttal**: elaborately designed experiments and detailed explanations to clarify the difference between the sampling strategies between ToRA&MARIO and DART;
> > > 2. **A4 & Follow-up Comment**: doubly checked clarification of our setting as pure distillation instead of self-KD;
> > > 3. **A2-3,5-7**: clarifications about the setting/details in the paper.
> > >
> > > Conducting the additional experiments within the limited rebuttal period is challenging. We would like to know whether our responses have addressed your concerns. If you still have other concerns, please give us an opportunity to clarify them.
> > >
> > > We sincerely hope that you can take a moment to reply, as it is very important for researchers and their efforts on this work.
> > >
> > > Best regards,
> > >
> > > The Authors

---

> ### Author Response · Authors · 2024-08-10
>
> Thanks for your reply! We would like to note that our work is not about **self** KD, but in general studying how to synthesize the best data from a strong model, which is like distilling from a stronger teacher model to student models, as we experimented with four different student models in the paper with the same teacher model. This setting aligns with many existing works [1,2,3,4,5,6,7], yet we replace their synthesis agent (GPT-4/3.5) with an open-weight model.
>
> Regarding the specific concern raised about using DeepSeekMath-7B-Base as the student model, this configuration is not strictly a self-KD setting either, because DeepSeekMath-7B-RL undergoes significant training with SFT+RL from DeepSeekMath-7B-Base on potentially large-scale human data,  positioning it more akin to a stronger teacher -> student distillation scenario, therefore, it is not surprising that the student is not significantly better than the stronger teacher, just as the previous works that distill from GPT-4 cannot surpass GPT-4. The self-KD case as in existing works [8,9,10,11] should correspond to `synthesizing from DeepSeekMath-7B-Base and training DeepSeekMath-7B-Base`, or `synthesizing from DeepSeekMath-7B-RL and training DeepSeekMath-7B-RL`. As the reviewer mentioned in the original review, we agree that these self-KD settings are reasonable and meaningful, we didn't adopt this setting simply because self-KD is not the focus of this paper, just as previous works that distill from GPT-4/3.5 [1,2,3,4,5,6,7] never explore the self-KD experiments with other synthesis agents.
>
> (A minor correction: the cited “86.7 GSM8K / 58.8 MATH” results by the reviewer is the tool-integrated results that rely on external tools and are not comparable, their CoT results are 88.2 GSM8K / 51.7 MATH).
>
> > the SFT model is comparable or even worse than the original DeepSeek-RL. So why not directly use the open-sourced DeepSeek-RL?
>
> High-quality synthetic data is of great values for the developers and open-source community, various developers may rely on those data to help strengthen their own models. **Our primary goal is not to create a math model for direct use, but to develop better data synthesis methods** — the roles of data synthesis and the synthetic data are not replaceable by “directly use the open-sourced DeepSeek-RL”, for example, imagine someone wants to boost the math ability of Mistral-7B during post-training while still keeping it as a generic model, they can utilize our approach to synthesize data from another math-specific model, and incorporate the data together with other SFT data as commonly practiced nowadays, but directly using DeepSeek-RL does not fulfill the goal.
>
> As the reviewer mentioned above, self-KD is one way to synthesize data for self-improvement, yet distilling from other models is very common as well, and our paper focuses on the later where student models do not need to surpass the teacher.
>
> [1] Luo, Haipeng, et al. "Wizardmath: Empowering mathematical reasoning for large language models via reinforced evol-instruct.” Preprint 2023.
>
> [2] Yu, Longhui, et al. "MetaMath: Bootstrap Your Own Mathematical Questions for Large Language Models." ICLR 2024.
>
> [3] Liu, Haoxiong, and Andrew Chi-Chih Yao. "Augmenting math word problems via iterative question composing." Preprint 2024.
>
> [4] Mitra, Arindam, et al. "Orca-math: Unlocking the potential of slms in grade school math.” Preprint 2024.
>
> [5] Li, Chen, et al. "Common 7b language models already possess strong math capabilities.” Preprint 2024.
>
> [6] Huang, Yiming, et al. "Key-point-driven data synthesis with its enhancement on mathematical reasoning." Preprint 2024.
>
> [7] Tang, Zhengyang, et al. "MathScale: Scaling Instruction Tuning for Mathematical Reasoning." ICML 2024.
>
> [8] Wang, Yizhong, et al. "Self-Instruct: Aligning Language Models with Self-Generated Instructions.” ACL 2023.
>
> [9] Dong, Hanze, et al. "RAFT: Reward rAnked FineTuning for Generative Foundation Model Alignment.” TMLR.
>
> [10] Yuan, Weizhe, et al. "Self-Rewarding Language Models.” ICML 2024.
>
> [11] Chen, Guoxin, et al. "AlphaMath Almost Zero: process Supervision without process.” Preprint 2024.

---

> ### Comment · Reviewer_quxz · 2024-08-12
>
> Thanks for your reminder that **DeepSeek-Math-RL** can achieve “86.7 GSM8K / 58.8 MATH” and "88.2 GSM8K / 51.7 MATH" for PoT and CoT, respectively.  As other partial concerns are addressed, I will increase rating from 3 to 4.  However, my main concern still exists based on the 3 following facts.
>
> 1. The SFT data is generated by running the **open-sourced DeepSeek-Math-RL**. In addition, the modified reject sampling based data generation method is not a significant innovation, which is widely used in previous works.
> 2. If **DeepSeek-Math-Base** is used as the base model, the performance of the fine-tuned model is only comparable with **the original DeepSeek-Math-RL**. The improvement is not significant.
> 3. If a less competitive base model is used, like **Llama** or **Mistral**, the performance of the fine-tuned models are significantly worse than the **original DeepSeek-Math-RL**.
>
> The standard reject sampling in the original DeepSeek paper already achieved strong results. In either way, I did not observe any advantage of using the proposed method over the **open-sourced DeepSeek-Math-RL**.

---

### Official Review · Reviewer_M9p3 · 2024-07-10

**Soundness:** 2
**Presentation:** 3
**Contribution:** 2
**Rating:** 4
**Confidence:** 4

**Summary:**

The paper introduces Difficulty-Aware Rejection Tuning (DART), a novel approach for enhancing the mathematical problem-solving capabilities of large language models (LLMs). Traditional methods often produce datasets biased towards easier queries, limiting the models' ability to learn from challenging examples. DART addresses this by allocating more sampling trials to difficult queries during the data synthesis phase. The authors created two strategies, Uniform and Prop2Diff, to ensure a balanced representation of easy and difficult queries. Using only open-weight models, the authors generated new, smaller datasets that prioritize difficult queries.

**Strengths:**

1. The DART method effectively addresses the bias towards easy queries in traditional rejection sampling, which is a significant contribution to the field.

2. The paper provides a thorough analysis of the biases in existing datasets and clearly explains how DART mitigates these issues.

3. The authors plan to make their datasets and models publicly available, contributing valuable resources to the research community.

**Weaknesses:**

1. The success of DART relies heavily on the ability of models to generate correct responses for difficult queries, which may not always be feasible for extremely challenging problems.

2. While the focus on difficult queries is commendable, the quality of the generated responses for these queries needs to be high to truly benefit the training process. The paper does not provide a detailed analysis of the quality of these responses.

3. The approach's reliance on extensive sampling for difficult queries might pose scalability issues, particularly for very large datasets or models with limited computational resources.

**Questions:**

1. How to be aware of difficulty if not labelled
2. How to choose $k_u$。
3. The details of Prop2Diff is missing. How many samples were generated for each difficulty level? What is the equation for generating numbers?

**Limitations:**

The limitation is mentioned in the conclusion section.

---

> ### Author Rebuttal · Authors · 2024-08-07
>
> Thanks for your positive comments! We address your concerns below.
>
> > **Q1**: The success of DART relies heavily on the ability of models to generate correct responses for difficult queries, which may not always be feasible for extremely challenging problems.
>
> **A1**: This is indeed a limitation for all rejection-sampling-based approaches, including both DART and the baselines. However, at least for the difficulty level of the MATH dataset, in Section 3.1 (Figure 2, Right) we have shown that DeepSeekMath-7B-RL is able to generate correct responses for nearly 100% of the queries in the MATH500 test set when sampling more.
>
> > **Q2**: While the focus on difficult queries is commendable, the quality of the generated responses for these queries needs to be high to truly benefit the training process. The paper does not provide a detailed analysis of the quality of these responses.
>
> A2: This is a good point. In this work, we just use the final answer to filter responses as in most previous works, while evaluating quality of these responses is non-trivial yet potentially beneficial -- an additional reward model may be needed for assessing their quality as in the Llama3 paper. We leave exploration of this for future work.
>
> > **Q3**: The approach's reliance on extensive sampling for difficult queries might pose scalability issues, particularly for very large datasets or models with limited computational resources.
>
> **A3**: We have conducted quantitative analysis about the synthesis cost in Section 4.3, please check it for details. While our method indeed requires extensive sampling for difficult queries, we highlight that the synthesis cost is one-time and can be amortized by distributing the obtained datasets to a large open-source community to use. Additionally, our synthesis agent is only a 7B-size model, unlike most other works that use GPT-4. Thus, our synthesis cost per example is much cheaper compared to previous works, although we do need to synthesize more examples.
>
> > **Q4**: How to be aware of difficulty if not labelled
>
> **A4**: We sample multiple responses for each query with DeepSeekMath-7B-RL and use the proportion of incorrect responses (i.e., fail rate) as the difficulty score. We have described this in Line 158-166 of the submission.
>
> > **Q5**: How to choose $k_u$
>
> **A5**:  Generally, according to the scaling curves in Figure 3,  larger $k_u$﻿ (or $k_p$﻿) correlates with higher accuracy. Thus, larger $k_u$﻿ (or $k_p$﻿) is overall preferred. However, the value of $k_u$﻿ (or $k_p$﻿) directly determines the final dataset size, so the choice of $k_u$﻿ (or $k_p$﻿) is primarily based on the desired final dataset size within resource constraints. In this paper, we choose $k_u=40$ and $k_p=192$ to ensure that both `DART-Math-Uniform` and `DART-Math-Hard` datasets end up with \~590K examples.
>
> > **Q6**: The details of Prop2Diff is missing. How many samples were generated for each difficulty level? What is the equation for generating numbers?
>
> **A6**: As we described in Line 137-139, we compute a difficulty score for each query (see Line 158-166) and the number of correct responses per query is linearly proportional to its difficulty score, where the most difficult query receives $k_p=192$ responses — suppose the difficulty score is $r_i\in[0,1]$, number of correct responses for a certain query $i$ is $k_p * r_i$. We keep sampling until the designated number of correct responses is collected or a max sampling cap $n_{\max}$ is reached (Line 145).

---

> > ### Comment · Reviewer_M9p3 · 2024-08-12
> >
> > While I appreciate the authors' efforts in addressing the feedback, I still have a few concerns that impact my overall assessment.
> >
> > 1. > "we have shown that DeepSeekMath-7B-RL is able to generate correct responses for nearly 100% of the queries in the MATH500 test set when sampling more".
> >
> > Even the most advanced closed-source models (GPT-4o or Claude-3) struggle to reach such high accuracy on this challenging dataset. It is unclear how increased sampling alone could lead to such an unprecedented level of accuracy. Thus, this result seems "suspicious", a further clarification on this point is necessary to ensure the results are credible.
> >
> > 2. > "we just use the final answer to filter responses as in most previous works,"
> >
> > However, this approach does not address the issue of "reward hacking," where models may generate the correct answer but fail to produce reasonable intermediate steps. This flaw significantly impacts the quality of the generated datasets and, by extension, the paper’s overall contribution. While I understand that the primary focus of this work is on data generation methods (as highlighted by the authors and other reviewers), the evaluation methodology used here undermines the reliability of the findings.

---

> > > ### Author Response · Authors · 2024-08-12
> > >
> > > Thanks for your reply!
> > >
> > > 1. The phenomenon that increased sampling would greatly improve pass@k accuracy on MATH dataset has been already observed by several previous works as well — for example, [1] reported over 80% MATH accuracy with GPT-4 when k=11, [2] reported 72% MATH accuracy with a weak Llama2-7B model when k=256, [3] reported over 85% MATH accuracy with DeepSeekMath-7B-RL when k=64. We totally understand that the reviewer thought these results were “suspicious”. To further clarify this, a high pass@k accuracy only means the correct answer exists among k responses, and it does not entail the model can solve the problem practically because it is hard to select the correct answer out. Additionally, pass@k accuracy is computed by matching the final answer, where the intermediate steps may be wrong even when the final answer is correct. Therefore, we think this high pass@k accuracy is understandable.
> > >
> > > 2. For the second point, we agree that the “reward hacking” phenomenon may indeed exist, and almost all rejection-sampling-based data synthesis methods including ours admit this limitation as it is non-trivial to guarantee the correctness of the intermediate steps. While imperfect with flaws, rejection-sampling-based synthetic data is still widely used [4, 5] to improve model’s final accuracy on these benchmarks, as we demonstrated in the paper as well — of course, the “reward hacking” phenomenon may be also present in benchmark evaluation where these models in previous works and our paper yield the correct final answer with wrong intermediate steps, yet how to evaluate math problem solving more faithfully considering intermediate steps is an evaluation problem beyond our scope and still an active research direction [6,7].
> > >
> > > [1] Toshniwal, Shubham, et al. "Openmathinstruct-1: A 1.8 million math instruction tuning dataset.” Preprint 2024.
> > >
> > > [2] Li, Chen, et al. "Common 7b language models already possess strong math capabilities.” Preprint 2024.
> > >
> > > [3] Shao, Zhihong, et al. "Deepseekmath: Pushing the limits of mathematical reasoning in open language models.” Preprint 2024.
> > >
> > > [4] Dubey, Abhimanyu, et al. "The Llama 3 Herd of Models.” Preprint 2024.
> > >
> > > [5] Yuan, Zheng, et al. "Scaling relationship on learning mathematical reasoning with large language models.” Preprint 2023.
> > >
> > > [6] Xia, Shijie, et al. "Evaluating Mathematical Reasoning Beyond Accuracy.” Preprint 2024.
> > >
> > > [7] Zeng, Zhongshen, et al. "MR-BEN: A Comprehensive Meta-Reasoning Benchmark for Large Language Models.” Preprint 2024.

---

> > > > ### Comment · Reviewer_M9p3 · 2024-08-12
> > > >
> > > > I thank the authors' response, but my concerns remain:
> > > >
> > > > 1. As the authors noted and as supported by the literature, achieving nearly 100% results by merely increasing sampling is challenging. This is because, regardless of the number of samples, the model may still produce incorrect answers due to its inherent limitations in solving the problem. This outcome aligns with the original performance reported by the DeepSeekMath model, where such results appear unrealistic.
> > > >
> > > > 2. The authors have acknowledged that "reward hacking" persists, which impacts the soundness of proposed method and datasets.
> > > >
> > > > Given these concerns, I believe the current paper is not yet ready for acceptance, and I have therefore lowered my score.

---

> > > > > ### Author Response · Authors · 2024-08-13
> > > > >
> > > > > Thank you for your feedback!
> > > > >
> > > > > 1. We want to clarify that it is reasonable to expect a high pass@k score and it does not contradict with the numbers from previous literature — all mentioned pass@k results above in the literature are from relatively small $k$, and they did not explore the effect when further increasing $k$. From the DeepSeekMath paper, particularly, their Figure 7 shows the pass@k accuracy trend is not close to convergence and still steadily improving when $k$ reaches 64 and pass@k is already over 85%, and they did not report results from larger $k$. **From the trend the DeepSeekMath paper reported in their Figure 7, a high pass@k like over 90% or 95% is actually quite expected when $k$ is further increased**, thus we don’t find our pass@k accuracy too unrealistic based on these previous studies. In fact, our results with smaller values of $k$ are consistent with them. For instance, as shown in Figure 2 (Right) of our paper, when $k\in[32,64]$, we achieve a pass@k rate of approximately 85-90%, which aligns well with DeepSeekMath's findings. Another point to help explain the high pass@k scores is that pass@k is an overestimated proxy of “complete correctness” that considers intermediate steps, due to the reward hacking issue the reviewer mentioned.
> > > > > 2. The reward hacking issue may exist not only in our method, but also in all rejection-sampling-based approaches for mathematical reasoning tasks. Synthetic datasets from rejection sampling are not completely correct and indeed contain noise due to reward hacking, yet they still help models empirically as evidenced and are being deployed practically in many works. Additionally, we want to note that "reward hacking" is still a difficult open problem, where little progress has been seen to evaluate intermediate steps faithfully — we feel further diving into reward hacking is an important direction for this field in general but a bit beyond the scope of this submission.

---

### Official Review · Reviewer_neCQ · 2024-07-12

**Soundness:** 2
**Presentation:** 3
**Contribution:** 2
**Rating:** 5
**Confidence:** 5

**Summary:**

The paper proposes a rejection sampling pipeline for automatically generating SFT data, emphasizing that harder data requires more trials. The difficulty is heuristically determined using the ratio of incorrect trials for each question. Experiments demonstrate that this method can outperform traditional rejection methods on various math benchmarks.

**Strengths:**

- The experiments are solid, showing significant improvements over traditional rejection methods.

- The paper is clearly written and easy to follow.

**Weaknesses:**

The proposed Prop2Diff strategy lacks innovation. Assigning more budget to more complex questions in data synthesis is a common practice. For instance, in [1], which successfully annotated 83.1% of MATH questions, it is evident that harder problems were allocated more budget in rejection sampling. [1] also indicates that fewer and harder data can significantly and efficiently improve performance. The authors should discuss the differences between their approach and the one used in [1] more thoroughly.

[1] ToRA: A Tool-Integrated Reasoning Agent for Mathematical Problem Solving

**Questions:**

Could you elaborate on how your approach differs from the rejection sampling strategy used in [1]?

[1] ToRA: A Tool-Integrated Reasoning Agent for Mathematical Problem Solving

**Limitations:**

Yes

---

> ### Author Rebuttal · Authors · 2024-08-07
>
> Thanks for your comments! We address your concerns below:
>
> > **Q1**: Assigning more budget to more complex questions in data synthesis is a common practice. For instance, in [1], which successfully annotated 83.1% of MATH questions
>
> **A1**: First, we note that most recent works in mathematical data synthesis, including nearly all commonly-used open-source datasets, allocate equal budgets to all questions regardless of their complexity [2,3,4,5,6]. While a few works such as the mentioned ToRA try assigning more budget to more complex questions to improve coverage, we highlight that ToRA still results in a bias towards easier queries. This is in stark contrast to our Uniform and Prop2diff strategies that focus on the distribution of responses, rather than merely improving coverage. Please see the Response to Q2 below for a detailed comparison with ToRA.
>
> Additionally, we would like to point out that 83.1% coverage mentioned from ToRA is not an exceptionally high number -- for comparison, the MetaMathQA-MATH-AnsAug dataset achieves 82.8% of coverage on the MATH training set with evenly allocated budgets yet still admits bias towards easy queries as analyzed in Figure 2 of our submission. Below we show the coverage rate per difficulty level of different approaches. The ToRA-Corpus-16k statistics show that it only covers 68% of the Level 5 MATH queries while `DART-Math` datasets cover 99.6%.
>
> | MATH training set coverage | Total | Level 1 | Level 2 | Level 3 | Level 4 | Level 5 |
> | --- | --- | --- | --- | --- | --- | --- |
> | ToRA-Corpus-16k-MATH | 83.1% | 97.7% | 91.6% | 86.5% | 81.3% | 68.0% |
> | MetaMath-MATH-AnsAug | 82.8% | 98.1% | 93.6% | 86.7% | 76.6% | 48.9% |
> | VRT Baseline | 84.9% | 99.6% | 98.2% | 95.2% | 89.8% | 62.9% |
> | `DART-Math-*` | 99.6% | 100% | 100% | 99.9% | 99.7% | 99.1% |
>
> > **Q2**: Could you elaborate on how your approach differs from the rejection sampling strategy used in [1]?
>
> **A2**: Please check the general rebuttal for a summary of difference of our approach from ToRA. The most important difference between our approach and ToRA is how responses are distributed across various queries -- While we adjust the distribution either to be uniform or to favor more difficult queries, ToRA focuses mainly on improving coverage without managing the distribution explicitly, leading to datasets that may still bias towards easier queries as shown in the results of the general author rebuttal.
>
> Here we supplement more details on how we replicate the ToRA synthesis pipeline to conduct the analysis present in the general author rebuttal. Below we show in the format as "ToRA's method -> how we adapt similar spirits for a simpler replication" step by step (we use CoT format rather than tool-integrated reasoning for a fairer comparison with our datasets):
>
> ---
> ---
>
> 1. Greedy decoding once for each problem in MATH&GSM8K with GPT-4, keeping the correct responses -> we follow this with GPT-4o mini.
> 2. Sampling for 10 trials for each problem not correctly answered by greedy decoding with GPT-4 and keeping up to 4 correct responses per problem (to form ToRA-Corpus-16k) -> we follow this with GPT-4o mini
> 3. Training CodeLlama models on ToRA-Corpus-16K to perform rejection sampling next -> to avoid additional training for a fairer comparison, we use DeepSeekMath-7B-RL to replace the trained CodeLLama models here to align with DART-Math
>     1. Sampling with 64 trials for each problem in MATH&GSK8K with CodeLlama, getting 233k distinct correct responses -> we follow this with DeepSeekMath-7B-RL, getting 733k distinct correct responses
>     2. correcting wrong responses by greedy decoding from the correct preceding portions (costing no more than 64 trials for each problem) with CodeLLaMA-34B, getting 69k corrected responses -> we simplify this by re-sampling another up to 64 trials per problem for all the incorrect responses, getting 225k correct samples.
>     3. Randomly selecting up to 4 correct responses per problem from 3.1&3.2 -> we exactly follow this.
> 4. Merge ToRA-Corpus-16k and data from step 3 to form the final training dataset of 69k responses -> we exactly follow this to form the final dataset of 72k responses.
>
> ---
> ---
>
> We show the average numbers of responses per problem for different difficulty levels and coverages on the MATH training set in the table of the general rebuttal — distribution-wise we can see that the ToRA pipeline will still produce fewer responses for difficult problems than the easy ones, while **`DART-Math-Hard` produces, for example, 10x more # of responses for MATH Level 5 questions than the GSM8K questions.**
>
> [1] Gou, Zhibin, et al. "Tora: A tool-integrated reasoning agent for mathematical problem solving." ICLR 2024.
>
> [2] Yu, Longhui, et al. "MetaMath: Bootstrap Your Own Mathematical Questions for Large Language Models." ICLR 2024.
>
> [3] Wang, Ke, et al. "MathCoder: Seamless Code Integration in LLMs for Enhanced Mathematical Reasoning." ICLR 2024.
>
> [4] Liu, Haoxiong, et al. "Augmenting math word problems via iterative question composing." Preprint 2024.
>
> [5] Huang, Yiming, et al. "Key-point-driven data synthesis with its enhancement on mathematical reasoning." Preprint 2024.
>
> [6] Tang, Zhengyang, et al. "MathScale: Scaling Instruction Tuning for Mathematical Reasoning." ICML 2024.

---

> ### Author Response · Authors · 2024-08-13
>
> Dear Reviewer neCQ,
>
> Sorry to disturb you for the last time, but only one day is left until the end of the reviewer-author discussion stage. We still do not know if you have received our newest response. To address your concerns, we wrote all the responses in detail and added new experiments to support them, including:
>
> 1. **A1**: showing 83.1% coverage mentioned from ToRA is not an exceptionally high number;
> 2. **A2 & Author Rebuttal**: elaborately designed experiments and detailed explanations to clarify the difference between the sampling strategies between ToRA and DART.
>
> Conducting the additional experiments within the limited rebuttal period is challenging. We would like to know whether our responses have addressed your concerns. If you still have other concerns, please give us an opportunity to clarify them.
>
> We sincerely hope that you can take a moment to reply, as it is very important for researchers and their efforts on this work.
>
> Best regards,
>
> The Authors

---

> > ### Comment · Reviewer_neCQ · 2024-08-13
> >
> > Thank you for your detailed responses, which have addressed most of my concerns. I'm pleased to raise my rating accordingly.

---

### Official Review · Reviewer_dSs8 · 2024-07-21

**Soundness:** 3
**Presentation:** 3
**Contribution:** 3
**Rating:** 6
**Confidence:** 4

**Summary:**

The paper presents an approach to improving the performance of LLMs in mathematical problem-solving. The authors identify that current datasets synthesized using proprietary models like GPT-4, are biased towards easier queries. To address this, they introduce Difficulty-Aware Rejection Tuning (DART), which allocates more trials to difficult queries during data synthesis. This method generates datasets focusing on difficult queries using an open-weight model, DeepSeekMath-7B-RL, without relying on proprietary models. The authors demonstrate that models fine-tuned on DART-Math datasets significantly those fine-tuned on traditional datasets across various mathematical benchmarks, and beat the best baseline by average of roughly 3-4%

**Strengths:**

- Technically solid paper with state-of-the-art results.
- Mostly well-presented and easy to understand.
- Comprehensive experiments and analysis.
- Decent impact in improving mathematical capabilities of LLMs, with the authors publicly releasing their dataset.
- By using an open-weight model, DeepSeekMath-7B-RL, the authors eliminate dependency on proprietary models like GPT-4, making the approach more accessible.

**Weaknesses:**

1. It is unclear how the hyperparameters of the baseline, VRT (vanilla rejection tuning), were tuned. For instance, as mentioned in Appendix A.2, sampling temperature is searched from 0.3 to 1.7 for DART. Was the same procedure used for VRT? Another caveat is the need for extensive hyperparameter tuning compared to baselines. Were similar extensive procedures for tuning performed for other baselines?
2. It is unclear if the improved performance of the proposed method is due to difficulty or the topic of the problem. For instance, LEVEL 5 Math problems may have a higher number of geometry questions (or at least their fail rate is higher, resulting in fewer samples in VRT). An analysis of topic-wise performance comparing DART and baseline methods may clarify this.

**Minor Weaknesses: **
1. It is unclear how much advantage the method would provide in the case of other multi-iteration fine-tuning methods such as STaR and v-STaR. For instance, it is possible that after multiple iterations, VRT performs similarly to DART, since a higher number of samples will be collected from even the hard problems in second or further iterations.
2. The data synthesis is only done using the DeepSeekMATH-7B model. It is unclear why this model was chosen. Previous methods using VRT-like methods typically use the same model for synthesis and generation. Thus, higher results in smaller models such as Llama-8B may partly be due to the use of stronger models' reasoning chains, making it similar to a distillation method.

**Questions:**

1. The authors use "fail rate" as a metric to measure difficulty. However, has any analysis been performed to measure how good an estimate it is of actual model accuracy?
2. In line 138, "we sample till correct responses for each query is proportional to its difficulty score," does it mean linearly proportional?
3. To the best of my knowledge, previous works usually use lower temperatures in the range of 0.7-1. However, the authors found 1.6 to be effective. Do the authors have a comparison of results between using a more standard temperature (e.g., 0.7 or 1) compared to 1.6?
4. In Line 253, the authors state: "We hypothesize that this is due to these models’ extensive pretraining on mathematical content." Do the authors have more points to substantiate this? For instance, it could be partly due to a slightly weaker or similar model being used to generate synthetic data. Further, the hypothesis: "This pretraining likely covers most skills that could be learned from the GSM8K and MATH training queries" may not be correct, since, at least for Llama2-70B, the model capacity should not be a bottleneck to achieving higher scores on MATH (e.g., Qwen models). Can the authors provide a more detailed reasoning behind this hypothesis?

**Limitations:**

Major Limitations are addressed in paper.

---

> ### Author Rebuttal · Authors · 2024-08-07
>
> Thank you for the positive comments! We address your concerns below.
>
> > **Q1**: It is unclear how the hyperparameters of the baseline, VRT, were tuned. For instance ……, sampling temperature is searched from 0.3 to 1.7 for DART.
>
> **A1**: We searched temperature from 0.3 to 1.7  according to accuracies by DeepSeekMath-7B-RL on the MATH training set, where $t=1.6$ is the highest temperature that does not suffer from a significant accuracy drop. $t=1.6$ is applied for both VRT and DART-Math. This choice should be fair since the parameter search is not specifically tailored for DART method.
>
> > **Q2**: It is unclear if the improved performance of the proposed method is due to difficulty or the topic of the problem. … An analysis of topic-wise performance comparing DART and baseline methods may clarify this.
>
> **A2**: Thanks for the advice! Difficulty and topic are naturally correlated. As shown in the following table for two models, both topic-wise and topic-macro-average scores (which assign equal weights to different topics) still show significant improvement on every topic by DART.
>
> | Model | Counting & Probability | Prealgebra | Number Theory | Intermediate Algebra | Algebra | Precalculus | Geometry | Micro Avg. | Macro Avg. |
> | --- | --- | --- | --- | --- | --- | --- | --- | --- | --- |
> | Llama3-8B-VRT | 34.2 | 57.8 | 30.7 | 20.4 | 59.6 | 22.5 | 29.0 | 39.7 | 36.3 |
> | `DART-Math-Llama3-8B` (Uniform) | 34.6 | **65.7** | 35.7 | 25.4 | 66.6 | 29.3 | 32.4 | 45.3 | 41.4 |
> | `DART-Math-Llama3-8B` (Prop2Diff) | **38.8** | 62.9 | **36.8** | **26.1** | **67.3** | **32.0** | **39.9** | **46.6** | **43.4** |
> | Mistral-7B-VRT | 32.1 | 56.3 | 29.6 | 19.0 | 58.4 | 22.2 | 30.7 | 38.7 | 35.5 |
> | `DART-Math-Mistral-7B` (Uniform) | 33.8 | 59.8 | 35.2 | 24.4 | 64.1 | 28.8 | 34.2 | 43.5 | 40.0 |
> | `DART-Math-Mistral-7B` (Prop2Diff) | **36.1** | **61.3** | **35.4** | **26.0** | **65.7** | **31.1** | **40.5** | **45.5** | **42.3** |
>
> > **Q3**: Another caveat is the need for extensive hyperparameter tuning compared to baselines
>
> **A3**: To clarify, we apply the same training hyperparameters to all the experiments on the same base models, thus there is no additional hyperparameter tuning for DART compared to the baselines during training. For data synthesis, the only additional hyperparameters specifically chosen for DART are # responses per query (Line 174-175) and the sampling cap (Line 145), which is mainly decided by the desired final dataset size considering our resource constraints, we did not  “tune” these synthesis hyperparameters according to final performance.
>
> > **Q4**: It is unclear how much advantage the method would provide in the case of other multi-iteration fine-tuning methods
>
> **A4**: This is a good point. DART is compatible with multi-iteration fine-tuning as well and we can use the DART strategy to manipulate data synthesis for each iteration. We leave exploration on this as future work.
>
> > **Q5**: The data synthesis is only done using the DeepSeekMATH-7B model. It is unclear why this model was chosen.
>
> **A5**: The DeepSeekMath-7B-RL model is chosen as the strongest open-source model (though with only 7B size) for mathematical problem-solving.
>
> > **Q6**: Previous methods using VRT-like methods typically use the same model for synthesis and generation. Thus, higher results in smaller models such as Llama-8B may partly be due to the use of stronger models' reasoning chains, making it similar to a distillation method.
>
> **A6**: We agree that our approach is similar to a distillation method, and our goal is to create the best synthetic data for mathematical problem-solving. The mentioned works that use the same model for synthesis represent a different line of work that pursues" self-improvement," while our paper can be viewed as pure data synthesis. This aligns with many existing works in this field that produced SOTA synthetic datasets using GPT-4, a stronger model.
>
> > **Q7**: In line 138, "we sample till correct responses for each query is proportional to its difficulty score," does it mean linearly proportional?
>
> **A7**: Yes, it means linearly proportional.
>
> > **Q8**: Do the authors have a comparison of results between using a more standard temperature (e.g., 0.7 or 1) compared to 1.6?
>
> **A8**: We didn’t compare temperatures in effects on the final performance, and we only compare temperatures in a preliminary stage by evaluating DeepSeekMath-7B-RL, as explained in A1 above.
>
> > **Q9**: In Line 253, the authors state: "We hypothesize that this is due to these models’ extensive pretraining on mathematical content." Do the authors have more points to substantiate this?
>
> **A9**: The hypothesis stems from the understanding that extended large-scale pretraining can reduce the need for meticulously curated SFT datasets. However, we did not rigorously verify this hypothesis, and we agree with the reviewer that "a slightly weaker or similar model being used to generate synthetic data" could be the reason as well.
>
> > **Q10**: the hypothesis: "This pretraining likely covers most skills that could be learned from the GSM8K and MATH training queries" may not be correct, since, at least for Llama2-70B, the model capacity should not be a bottleneck to achieving higher scores
>
> **A10**: This is an interesting point. To clarify, we didn't mean model capacity is a bottleneck, instead, we meant the training queries are the bottleneck --  the queries in `DART-Math` are simply the original GSM8K & MATH training queries that limit the scope that the final dataset can generalize to. Therefore, the generalization accuracy could be bottlenecked by the training query scope no matter how we improve the synthetic responses. Our hypothesis is that math-specific pretraining may already cover most skills that could be learned from these queries, and further training with those queries could not provide significant values. A potential future direction could be synthesizing new queries.

---

### Author Rebuttal · Authors · 2024-08-07

We thank all the reviewers for the insightful comments! While we address most concerns in the individual rebuttals, here In the general rebuttal we would like to clarify the difference between our approach and ToRA [1] / MARIO [2], a concern raised by Reviewer neCQ and Reviewer quxz.

The most important difference between our approach and ToRA/MARIO is how responses are distributed across various queries -- While we adjust the distribution either to be uniform or to favor more difficult queries, **ToRA/MARIO focus mainly on improving coverage without managing the distribution explicitly, leading to datasets that may still bias towards easier queries** (as we show below). As demonstrated in Figure 4 (Left) of our submission, a high coverage rate (VRT+Cover) alone does not guarantee superior performance. Since both ToRA and MARIO have not released their datasets, we replicate a version of their synthesis pipelines that is comparable with DART to illustrate that the resultant datasets still bias towards easier queries.

We detail our specific replication process for ToRA and MARIO in the individual rebuttals to Reviewer neCQ and Reviewer quxz respectively. Below we show the average number of responses per problem for different difficulty levels below of our replicated ToRA and MARIO datasets and the DART-Math datasets. Our replicated ToRA and MARIO end up with a similar MATH coverage ratio and total size to their original ones. While the absolute number of responses is not directly comparable between different methods, distribution-wise we can see that ToRA/MARIO still produce fewer responses for difficult problems than the easy ones. This contrasts sharply with `DART-MATH-Hard`, which produces, for example, 10x more responses for the MATH Level 5 queries than for the GSM8K queries.

|  | GSM8K | MATH Level 1 | MATH Level 2 | MATH Level 3 | MATH Level 4 | MATH Level 5 | Our Size (Original Size in Their Papers) | MATH/Train Coverage |
| --- | --- | --- | --- | --- | --- | --- | --- | --- |
| ToRA | 5.03 | 5.01 | 4.99 | 4.95 | 4.77 | 3.84 | 72k (69k) | 93.4% |
| MARIO | 2.02 | 2.01 | 1.98 | 1.94 | 1.89 | 1.57 | 29k (29k) | 91.3%  |
| `DART-Math-Uniform` | 39.93 | 40.00 | 40.00 | 39.80 | 39.54 | 37.14 | 585k | 99.6% |
| `DART-Math-Hard` | 8.49 | 14.28 | 33.52 | 54.94 | 79.59 | 107.06 | 590k | 99.6% |

We hope the response above clarifies the difference between ToRA/MARIO and our approach.

[1] Gou, Zhibin, et al. "Tora: A tool-integrated reasoning agent for mathematical problem solving." ICLR 2024.

[2] Liao, Minpeng, et al. "MARIO: MAth Reasoning with code Interpreter Output--A Reproducible Pipeline." arXiv 2024.

---

### Author Response · Authors · 2024-08-12
**Kind Reminder before the End of Discussion**

Dear Reviewers,

We sincerely appreciate your feedback. Now, we would like to send you a friendly reminder that the discussion stage will be complete soon. If you have any additional concerns or questions, please do not hesitate to let us know. We are more than willing to discuss any further suggestions or issues you might have.

Best regards,

The Authors

---

### Decision · Program_Chairs · 2024-09-25

**Decision:**

Accept (poster)

**Comment:**

The paper is about mathematical reasoning. First, the authors find that there are synthetic datasets synthesized by general LLMs like GPT4 that result in a dataset biased towards easier questions. In this paper, the authors present a different approach based on DeepSeek-Math that results in a more balanced dataset with difficult questions well represented.

The paper received a 3, 4, 4, 7, but the 3 increased to 4, 4 increased to 5, and 7 reduced the score to 6 based on other discussion. So, with a 4-4-5-6, the paper is quite borderline. There are two main criticisms of the paper. First, as one of the critical reviewers pointed out, and did make the highly positive reviewer go milder... concerns about a specific evlauation. Basically, many other papers used general models like GPT4 which are weaker in mathematical reasoning for dataset geneartion. This paper's methodology, when used over same models, also results in a weaker synthetic dataset. On the other hand, when DeepSeek-Math-RL is used to generate the dataset, the dataset indeed is much better than before, but it is only better than other other GPT4-based datasets. When comparing this dataset, the authors fine-tune DeepSeek-Math-"Base" model on this dataset, the performance is similar to (or sometimes worse than) the original DeepSeek-Math-RL. In other words, if we already had DeepSeek-Math-RL model at our disposal (since it was used for dataset generation), it is not clear why not use this model directly and unsupervised, and why fine-tune another model to reach this model's performance using their data.

The authors, in response, argue that "Our primary goal is not to create a math model for direct use, but to develop better data synthesis methods — the roles of data synthesis and the synthetic data are not replaceable by “directly use the open-sourced DeepSeek-RL”, for example, imagine someone wants to boost the math ability of Mistral-7B during post-training while still keeping it as a generic model, they can utilize our approach to synthesize data from another math-specific model, and incorporate the data together with other SFT data as commonly practiced nowadays, but directly using DeepSeek-RL does not fulfill the goal."

As a meta-reviewer, I understand both points of view, and don't side with either. A better method for dataset generation is always a good idea. But, is the new dataset better because of a stronger LLM generating the data (DeepSeek-Math-RL) or a better algorithm is not clear. For a rigorous study, evaluation could have been done by using different LLMs to generate data, and different base models for fine tuning, creating a stronger understanding of values of various components. At the same time, creating a stronger dataset is indeed valuable for the reasons authors point out -- because it can help train all models, and not just rely on one model.

The other criticism of reviewers is wrt comparison of the algorithm vs MARIO and ToRA -- two other approaches for dataset creation. The reviewer believes that the technical novelty over these two is incremental. In response, authors point out that "ToRA&MARIO focus mainly on improving coverage without managing the distribution explicitly, leading to datasets that may still bias towards easier queries as we show quantitatively". The authors show with detailed experimental results that ToRA/MARIO produce fewer responses for difficult problems than the easy ones. I believe this experiment sufficiently addressed this concern.

Finally, there are some concerns about whether the model leads to "reward hacking", since the paper uses the final answer to filter responses as in most previous works. The reviewer considers it a super important phenomena strong enough to reject the paper, but I do not think so. This is true for all papers that use this kind of synthetic data generation, and as long as performance improves, this is still valuable to study. And the reviewer also pointed out the high pass@k accuracy for which I am comfortable with the authors answer.

To summarize, the only one criticism that continues to linger in my mind, is the use of different LLMs for data generation and training and comparing it to other datasets generated by a different LLM. They are consistently comparing their method with other open-source datasets created with a different model. A more rigorous experimental study would have made the paper a clearer accept. Still, the method does show improvements compared to vanilla rejection tuning, which is commonly used. With what the authors have shown in experiments, the scores are consistently better across settings. So, I will say that the paper is a borderline accept -- it can be taken in if there is space... but it would not be wrong, if the paper is rejected and the authors are asked to make the evaluation stronger.